# Major and Trace-Element Chemistry of Cr-Spinel in Upper Mantle Xenoliths from East Antarctica

**Alexandre V. Andronikov *** , **Irina E. Andronikova and Ondrej Pour**

Department of Geochemistry and Laboratories, Czech Geological Survey, Geologicka 6,
15200 Prague, Czech Republic; irina.andronikova@geology.cz (I.E.A.); ondrej.pour@geology.cz (O.P.)
* Correspondence: alexandre.andronikov@geology.cz

**Abstract:** Cr-spinels in the upper mantle peridotite xenoliths from two Late Mesozoic intrusions of alkaline-ultramafic rocks in Jetty Peninsula (East Antarctica) were studied in situ for their major and trace-element compositions by SEM and LA-ICP-MS. The upper mantle xenoliths were collected from the magmatic bodies "sampled" from different upper mantle domains. One domain was represented by mostly lherzolites (Cpx-poor Spl, Cpx-rich Spl and Spl-Grt) and another one by Spl harzburgites and dunites. Spinels occur as grains of different shapes, sizes and origins. Three main textural types of spinel were identified: primary spinel represented by clean homogeneous grains, a rim of recrystallization/resorption surrounding primary spinel grains and irregular interstitial resorbed grains. Primary spinels are characterized by the concentrations of $Al_2O_3$ 21–51 wt%, MgO 15–20 wt%, FeO 10–24 wt% and $Cr_2O_3$ 14–37 wt% with the Cr# of 0.16–0.54. Most trace elements are present in spinels in very low amounts. Only Ti, V, Mn, Co, Ni, Zn and Ga display concentrations in the range of tens to hundreds (up to thousands) ppm. Concentrations of other trace elements vary from below the detection limit to <10 ppm. Spinel major oxide and trace element features allowed the suggestion that the studied upper mantle peridotites represent both simple melt residues and residues strongly influenced by the MORB-like and the SSZ-related melts. The MORB-like melts may be related to the beginning of the Lambert–Amery rift system development, whereas SSZ-like melts could be related to reactivation of SSZ material buried during much earlier amalgamation of East Antarctica.

**Keywords:** spinel; trace-elements; upper mantle xenoliths; East Antarctica; SEM; LA-ICP-MS

## 1. Introduction

Small Late Jurassic to Early Cretaceous intrusions of alkaline-ultramafic rocks in Jetty Peninsula (northern Prince Charles Mountains; East Antarctica; Figure 1) contain multiple upper mantle xenoliths [1–7]. These xenoliths, derived from the subcontinental lithospheric mantle (SCLM), carry information about mineral and geochemical changes in the mantle. The xenoliths in Jetty Peninsula were mostly found in two intrusive bodies: Yuzhnoe (southern) and Severnoe (northern) located some 20 km apart. The upper mantle xenoliths in the Yuzhnoe body are represented by lherzolite of three main groups: (i) clinopyroxene (Cpx)-poor spinel (Spl) lherzolites, (ii) Cpx-rich Spl lherzolites and (iii) Spl-garnet (Grt) lherzolites [3–5]. The upper mantle xenoliths in the Severnoe body are represented by mainly (i) Spl-harzburgites and (ii) Spl-dunites [3,8].

The geological settings and tectonic characteristics of the region have been described in detail in a number of publications (e.g., [9–12] and references therein). Briefly, the studied area lies on the flank of the small rift system formed ca. 650 Ma and presently is located on the western side of the larger Lambert–Amery rift system which was active from the Mesozoic to Cenozoic [6,7]. Tectonically, amalgamation of East Antarctica is believed to have taken place either ca. 1400–1000 Ma through arc accretion and collision processes or ca. 580–500 Ma through continental collision (see [6,13] and references therein). In either scenario, the continental crust and the SCLM beneath the region are not homogeneous [14,15]. The difference in composition and lithology of the xenoliths found in the

two intrusive bodies suggests that the ascending magmas may have sampled two different upper mantle domains (see [8]). The deepest-sampled mantle rocks were found in the Yuzhnoe intrusion [2,4,16,17]. The Severnoe intrusion sampled overall shallower upper mantle levels, represented by the rocks being petrographically different from those sampled by the Yuzhnoe intrusion [3,4].

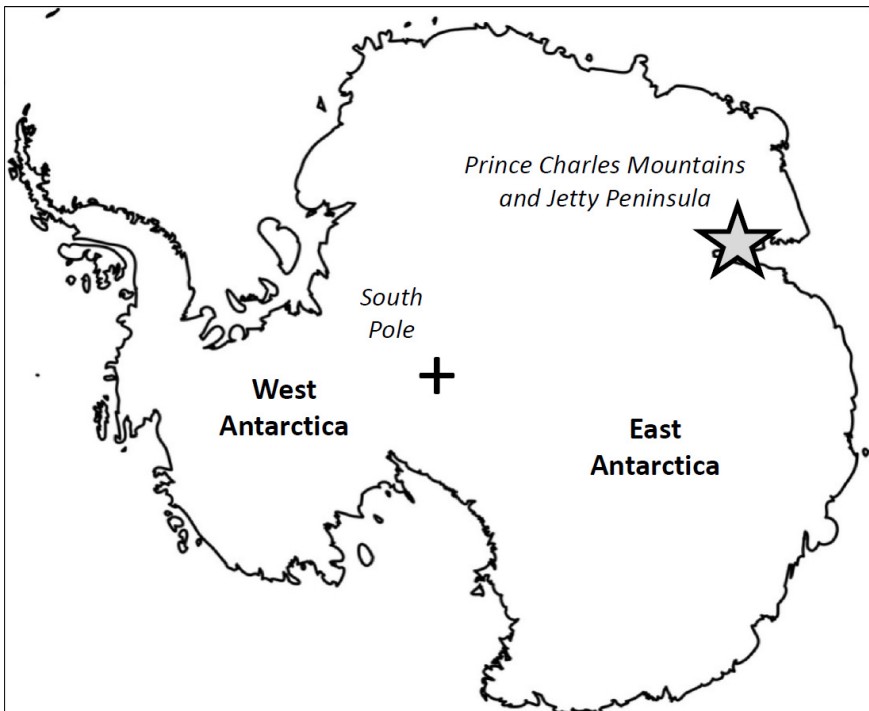

**Figure 1.** Schematic map of Antarctica showing the position of the Prince Charles Mountains and Jetty Peninsula (star). Detailed information on the studied intrusive xenolith-bearing bodies location is given in [4,6].

Spinel is a common phase in mantle rocks and is mostly represented by its Cr-rich variety (e.g., [18–20] and references therein). Spinel is volumetrically a minor mineral in the upper mantle xenoliths, but it is one of the most stable mantle minerals resistant to the secondary alterations, particularly when compared to other upper mantle minerals [18–23]. However, Cr-spinel can be modified by cryptic metasomatism through fluid (melt) to rock interaction [19,22,24]. That allows spinel to provide information on processes of both depletion and enrichment in the upper mantle. Although spinel in mantle peridotite xenoliths from different localities worldwide has been extensively studied to trace P-T-$f$O$_2$ conditions ([8,25,26] and references therein), studies of trace elements in the upper mantle spinel are very scarce (e.g., [22,24,27,28]). To the best of our knowledge, only one work so far has dealt with trace element composition of spinels in Antarctic upper mantle xenoliths (Victoria Land, West Antarctic [29]).

Major oxide composition of spinel in the upper mantle xenoliths from East Antarctica was documented in several works (e.g., [3,4,8]). However, trace element geochemistry of the mineral is completely unstudied yet. We conducted in situ scanning electron microscopy (SEM) and laser ablation inductively coupled plasma-mass spectrometry (LA-ICP-MS) analyses of spinel in peridotite xenoliths from Jetty Peninsula in order to constrain mantle processes such as melting and mantle rock to melt interaction. The present work was based on the xenolith samples previously studied for bulk rock, rock-forming mineral and sulfide geochemistry [2–4,7,16].

## 2. Materials and Methods

### 2.1. Brief Characteristics of the Xenoliths

The Cpx-poor Spl lherzolites are medium to coarse-grained inequigranular rocks composed of 78–82% olivine (Ol), 10–14% orthopyroxene (Opx), 4–7% Cpx, 1.5–5% Spl and insignificant amounts of carbonates (Carb) and sulfides (Sulf). Low concentrations of CaO and $Al_2O_3$ (0.8–1.5 wt% and 0.5–1.6 wt%, respectively) and high Mg# [100 Mg/(Mg + Fe) (at%)] of 90.2 to 91.2 suggest depletion by partial melting [3,4]. The rocks also experienced slight metasomatic enrichment after significant depletion (see [3–5].

The Cpx-rich Spl lherzolites are coarse-grained intergranular rocks consisting of 66–75% Ol, 11–16% Opx, 9–14% Cpx and 1–4% Spl (±Carb and Sulf). Most Cpx and Spl grains display recrystallized edges resulted from melt infiltration [4,30]. Recrystallized edges contain melt pockets produced by either the mineral melting or melt percolation along the grain boundaries [4,30]. Although the rocks are less depleted than the Cpx-poor lherzolites (Mg# 86.8–90.9; CaO 2.0–3.3 wt%; $Al_2O_3$ 2.2–3.0 wt%), they also show history of both depletion and later metasomatic enrichment [3–5].

The Spl-Grt lherzolites are coarse-intergranular, partly porphyroclastic rocks consisting of 57–66% Ol, 18–23% Opx, 11–16% Cpx, 2–7% Grt and 0.2–0.5% Spl (±Carb and Sulf). The bulk rock compositions are characterized by Mg# of 89.0–91.2, CaO of 2.3–3.4 wt% and $Al_2O_3$ of 2.3–4.1 wt%. Chemical composition of the rocks suggests that they experienced metasomatic overprints similar to those for Cpx-rich Spl lherzolites [4,5].

The Spl harzburgites are coarse-grained rocks with hypidiomorphic-granular to porphyroblastic textures composed of 75–90% Ol, 10–20% Opx, 0.5–2% Spl and insignificant amounts of Cpx, magnetite, Sulf and Carb. The bulk rock composition is characterized by high Mg# (90.2–90.6) and low CaO of 1.1–1.3 wt% and $Al_2O_3$ (0.9–1.1 wt%). A significant metasomatic enrichment after initial depletion of the rocks by the partial melting was suggested for harzburgites [3].

The dunites are coarse-grained granular rocks consisting of 85–99% Ol, 1–3% Opx, very variable (1–15%) amounts of Spl and insignificant amounts of magnetite, Carb and phlogopite. Dunites display a slightly lower Mg# of 86.5–88.9 than harzburgites at similar to lower $Al_2O_3$ (0.3–1.1 wt%) and lower CaO (0.6–1.1 wt%) concentrations. That, along with the rare Earth elements (REE) bulk rock patterns, suggests initial significant depletion of the rocks by the partial melting followed by strong metasomatic enrichment [3] which is, in particular, pronounced in the occasional presence of phlogopite.

Chemical composition of the rock-forming minerals in the upper mantle xenoliths from Jetty Peninsula is described in detail in [2–4]. Overall, Ol in the Cpx-poor Spl lherzolites is Mg-rich (Mg# 90.7–91.4) and uniform in composition as well as Opx, which is Al-poor (2.4–3.8 wt% $Al_2O_3$). Clinopyroxene-poor Spl lherzolites have only one generation of Cpx, whereas the Cpx-rich Spl lherzolites display Cpx of two generations varying from compositions similar to those in the Cpx-poor lherzolites to more Fe-rich compositions. Garnet in Spl-Grt lherzolites is pyrope-rich (Mg# 82.1–86.8), with 1.1–2.0 wt% $Cr_2O_3$ and 1.3–5.6 wt% CaO. Spinel displays the highest Cr# [Cr/(Cr + Al) (wt%)] in the Cpx-poor Spl lherzolites (0.26–0.35) and the lowest in the Spl-Grt rocks (0.14–0.23) among the lherzolite xenoliths. Harzburgites are characterized by slightly less Mg Ol (Mg# 88.4–88.6), Mg-richer Opx (Mg# 91.8–92.9) with high concentrations of Al (2.5–2.7 wt% $Al_2O_3$) and moderately Cr-rich Spl (Cr# 0.20–0.44). Chemical composition of Ol in dunites is similar to that in harzburgites (Mg# of 87.8–88.9), but spinels are much richer in Cr (Cr# = 0.43–0.66).

The Jetty Peninsula upper mantle xenoliths display evidence for multiple episodes of metasomatic enrichment through silicate, carbonate and sulfide melts [2,4,7,16,17]. The existence of such melts is confirmed by the presence of glasses along grain boundaries, inside resorbed Cpx and Spl rims and as interstitial patches and interstitial carbonate patches and small (<10 μm) sulfide blebs occurring as inclusions in spinels.

The studied lherzolite xenoliths display the ambient temperatures ranging from 1100–1135 °C for Spl-Grt lherzolites (for pressures up to 24 kbar that corresponds P-T conditions at depths of 75–80 km), to 1090–1100 °C for Cpx-rich Spl lherzolites and to

850–1035 °C for Cpx-poor Spl lherzolites [4]. Garnet-free xenoliths were trapped from the shallower upper mantle horizons (as shallow as the mantle to crust boundary). Non-lherzolite xenoliths display the ambient temperatures of 975–1025 °C for harzburgites and 815–890 °C for dunites at suggested pressure of 12–19 kbar [3,4,8]. Oxygen fugacity ($fO_2$) relative to the FMQ buffer (Δlog(FMQ) lies between −0.17 and −2.62 (mean −1.03) for Cpx-poor Spl lherzolites, between +0.15 and −2.13 (mean −0.35) for Cpx-rich Spl lherzolites, between −0.22 and −1.18 (mean −0.75) for Spl-Grt lherzolites, between +0.23 and +0.97 (mean +0.60) for harzburgites and between +0.66 and +0.82 (mean +0.74) for dunites [4,8].

The seven selected lherzolite samples from the southern SCLM domain vary from the Cpx-poor Spl lherzolites (U-3/4-2, U-1/4-3) to the Cpx-rich Spl lherzolites (UN-1, XLT-5) and to the Spl-Grt lherzolites (DK-8/3, DN-1, DN-4). Two samples of Spl harzburgite (SN-A30, SN-N18) and two samples of dunite (SN-N6, SN-N9) represent the rocks from the second (northern) upper mantle domain.

### 2.2. Analytical Techniques

Polished sample mounts of 25 mm diameter were prepared from the xenolith chips for in situ analyses. Spinels were first identified and studied visually in the reflected light with the use of the Carl Zeiss MicroImaging GmbH microscope equipped with the Axio Imager 2. Then, spinels were analyzed for major and trace element concentrations with the use of the SEM and LA-ICP-MS methods at the Department of Geochemistry and Laboratories of the Czech Geological Survey (Prague, Czech Republic). As a rule, we analyzed two spots per sample: at the center of the grain and at its edge (rim).

### 2.2.1. SEM Analyses

The SEM analyses were performed in order to allow selection of grains for further LA-ICP-MS runs and to provide the necessary major element concentrations for quantitative spinel analysis. Analyses of major elements were performed by using a Tescan MIRA3 GMU FEG-SEM equipped with a wavelength-dispersive spectroscopy (WDS) microanalyzer (Oxford Instruments Wave; UK). The analyses were conducted with the accelerating voltage of 15 kV, beam current of 8 nA and working distance of 15 mm. Counting times were 20 s for the peak position and 10 s for background positions. The elements of routine analyses of spinels were Si, Ti, Al, Cr, Fe, Mn, Mg, Ni, Ca, Na and K. The following standards were used for quantitative analyses: jadeite (for Si and Na), chromite (Al, Cr, Fe and Mg), rutile (Ti), rhodonite (Mn), orthoclase (K), diopside (Ca) and pure Ni. Analyses were processed by the INCA software (Oxford Instruments). Concentrations of ferric iron ($Fe^{3+}$) in spinel were calculated with the use of the equation from [31] based on stoichiometric criteria.

### 2.2.2. LA-ICP-MS Analyses

The applied LA-ICP-MS analytical system consisted of the Agilent 7900 quadrupole (Q)-ICP-MS (Agilent Technologies Inc., Santa Clara, CA USA) coupled with the Analyte Excite Excimer 193 nm LA system (Photon Machines, Redmond, WA, USA) equipped with the two-volume HelEx ablation cell. The laser gas system included nitrogen for purging of the laser beam path, helium carrier gas, internal premix argon-fluoride and the flush helium gas. The Q-ICP-MS was equipped with standard Ni sampler and skimmer cones and a quartz torch with a 2.5 mm glass injector. The laser beam diameter (a spot size) varied from 25 to 65 μm (depending on the analyzed grain size). The applied laser fluence was 3.92–4.71 J/cm$^2$ at the laser pulse rate of 10 Hz. A typical analysis consisted of 20 s of background acquisition (gas blank), 30–60 s of sample ablation and 30–40 s of wash-out time (Figure 2). The GLITTER 3.0 software package was used as a data reduction program [32,33]. An internal standardization was based on iron concentrations determined by the *SEM* analysis (as FeO$_{tot}$). Three reference materials were used for the external calibration: a USGS GSE-2G standard (basaltic glass; [34] to calibrate V, Mn, Co, Ni, Zn and Sr, a NIST-610 synthetic glass standard [35] to calibrate Se, Te, Pt, Au and Tl and the USGS GSD-2G standard (basaltic glass; [34]) to calibrate the rest of the elements. All standards

were analyzed at the beginning and the end of each analytical session and also after every 10–15 individual measurements in order to monitor sensitivity drift. The following isotopes were monitored during the analyses: $^{45}Sc$, $^{47}Ti$, $^{51}V$, $^{55}Mn$, $^{57}Fe$, $^{59}Co$, $^{61}Ni$, $^{65}Cu$, $^{66}Zn$, $^{71}Ga$, $^{77}Se$, $^{85}Rb$, $^{88}Sr$, $^{89}Y$, $^{93}Nb$, $^{94}Zr$, $^{107}Ag$, $^{111}Cd$, $^{115}In$, $^{118}Sn$, $^{121}Sb$, $^{125}Te$, $^{137}Ba$, $^{139}La$, $^{140}Ce$, $^{141}Pr$, $^{146}Nd$, $^{147}Sm$, $^{153}Eu$, $^{157}Gd$, $^{159}Tb$, $^{163}Dy$, $^{165}Ho$, $^{166}Er$, $^{169}Tm$, $^{172}Yb$, $^{175}Lu$, $^{178}Hf$, $^{181}Ta$, $^{195}Pt$, $^{197}Au$, $^{205}Tl$, $^{208}Pb$ and $^{209}Bi$. Element concentrations were calculated from the obtained isotope signal intensities.

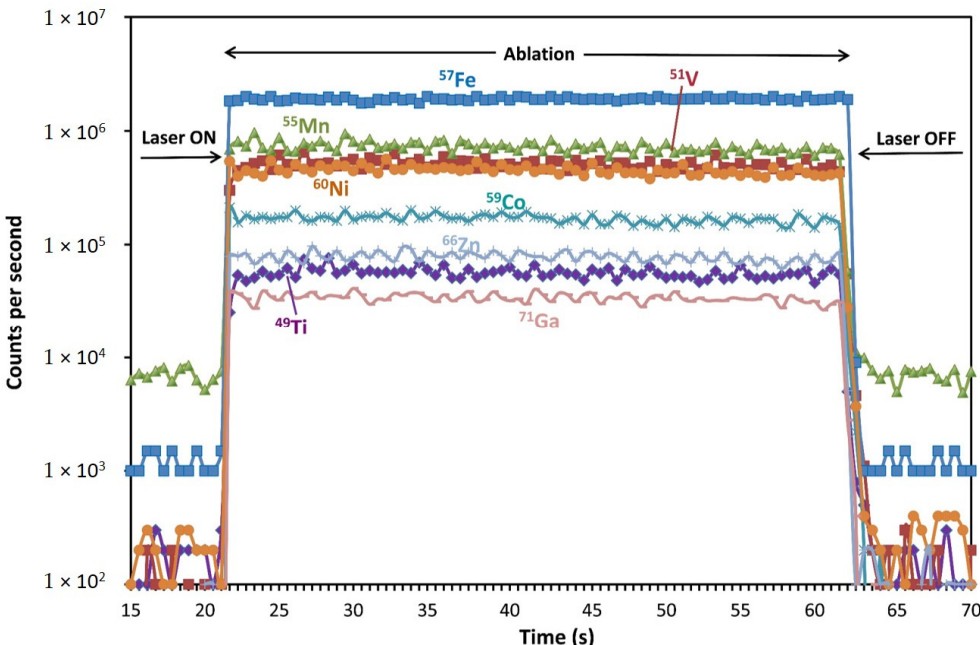

**Figure 2.** Representative time-resolved LA-ICP-MS depth profiles for a spinel (logarithmic scale) with a few selected isotopes shown (primary Cr-spinel from Spl-Grt lherzolite sample DK-8/3; beam diameter is 65 μm). From the left, the background count is 23 s (not shown completely), followed by 42 s of the ablation time, which is integrated and by 40 s of the wash-out time (not shown completely).

## 3. Results

### 3.1. Spinel Petrography and Major-Element Composition

Spinels in the Jetty Peninsula xenoliths occur as grains of different shape, size and origin. We grouped spinels into four main types according to their textural occurrence. Primary spinel (i) is located between grains of other primary xenolith minerals (Ol, Opx, Cpx) and is represented by xenomorphic grains of different sizes. The following two spinel types are developed after the primary spinel: either a rim (ii) around primary spinel grains or completely resorbed irregular grains (iii). The last spinel type identified here is represented by small grains inside a kelyphite rim replacing garnet (iv). Modal abundance of spinel in the studied xenoliths is as follows: dunite > Cpx-poor lherzolite ≥ Cpx-rich lherzolite ≥ harzburgite > Spl-Grt lherzolite. Major element characteristics of the analyzed spinels are given in Tables 1 and 2 and in the Electronic Supplementary Table S1.

#### 3.1.1. Spinel in Cpx-Poor Spl Lherzolites

Spinels occur here as sub-idiomorphic grains of 20–500 μm size scattered unevenly through the rock. Three types of spinel were identified: primary spinel represented by clean homogeneous grains (Figure 3a), a thin (5–20 μm) sporadic rim of secondary spinel crystals (Spl-2 in Figure 3b) along the edges of primary spinel (Spl-1 in Figure 3b) and rare small oval spinel inclusions in primary silicate minerals, in which spinels were not analyzed because of their size (Figure 3a). In the xenoliths U-3/4-2, spinel is present only as rim-free primary grains.

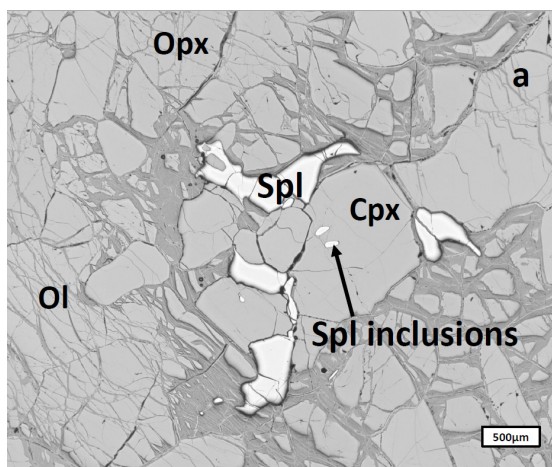 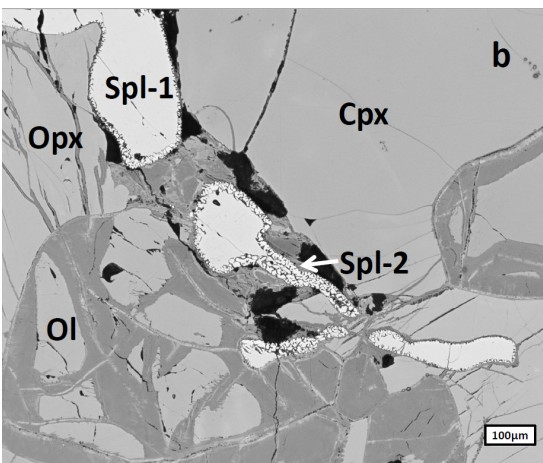

**Figure 3.** Types of spinels from Cpx-poor Spl lherzolites: (**a**), Primary spinel represented by clean homogeneous grains (Spl) and small oval spinel inclusions (Spl inclusions) in primary clinopyroxene; (**b**), Primary spinel grains (Spl-1) surrounded by a sporadic thin rim of secondary spinel (Spl-2). Cpx, clinopyroxene; Ol, olivine; Opx, orthopyroxene.

Compositionally, primary spinels are represented by the solid solution of the spinel (71.9–73.2%) and chromite (23.3–26.1%) components with insignificant amounts of the magnetite component. Chemically, such spinels are almost Ti-free (<0.01 wt% $TiO_2$) and are characterized by the high amounts of Al (42.4–43.8 wt% $Al_2O_3$) and Mg (18.9–20.2 wt% MgO) and moderate amounts of Fe (10.3–11.8 wt% FeO) and Cr (24.3–26.3 wt% $Cr_2O_3$) with the Cr# varying from 0.27 to 0.29.

A secondary rim along the edges of primary spinel grain is represented by the solid solution of the spinel component (66.5–66.6%), the chromite component (30.0–30.1%) and insignificant amounts of the magnetite and ulvöspinel components. Chemically, the rim spinels are characterized by low but measurable amounts of Ti (0.2–0.3 wt% $TiO_2$), lower amounts of Al (34.7–35.8 wt% $Al_2O_3$) and Mg (18.7–19.0 wt% MgO), similar to the host primary spinel amounts of Fe (11.1–11.6 wt% FeO) and significantly higher Cr amounts (31.8–32.7 wt% $Cr_2O_3$) with the Cr# of 0.37–0.39.

### 3.1.2. Spinel in Cpx-Rich Spl Lherzolites

Spinels occur here as irregular interstitial grains of 20–1000 μm size scattered through the rock's body. Two types of spinel were identified: primary spinel represented by clean homogeneous grains of irregular shape (Figure 4a,b) and a thin (5–20 μm) rim of the secondary spinel (Spl-2 in Figure 4c) sporadically surrounding primary spinel grains (Spl-1 in Figure 4c).

Compositionally, primary spinel is represented by the solid solution of the spinel component (79.3–80.2%), the chromite component (16.4–17.6%) and insignificant amounts of the magnetite and ulvöspinel components. Chemically, primary spinels are characterized by moderate amounts of Ti (0.2–0.3 wt% $TiO_2$), high amounts of Al (45.3–51.7 wt% $Al_2O_3$) and Mg (19.4–21.1 wt% MgO) and moderately low amounts of Fe (11.4–13.2 wt% FeO) and Cr (14.9–20.1 wt% $Cr_2O_3$) with the Cr# varying from 0.16 to 0.23.

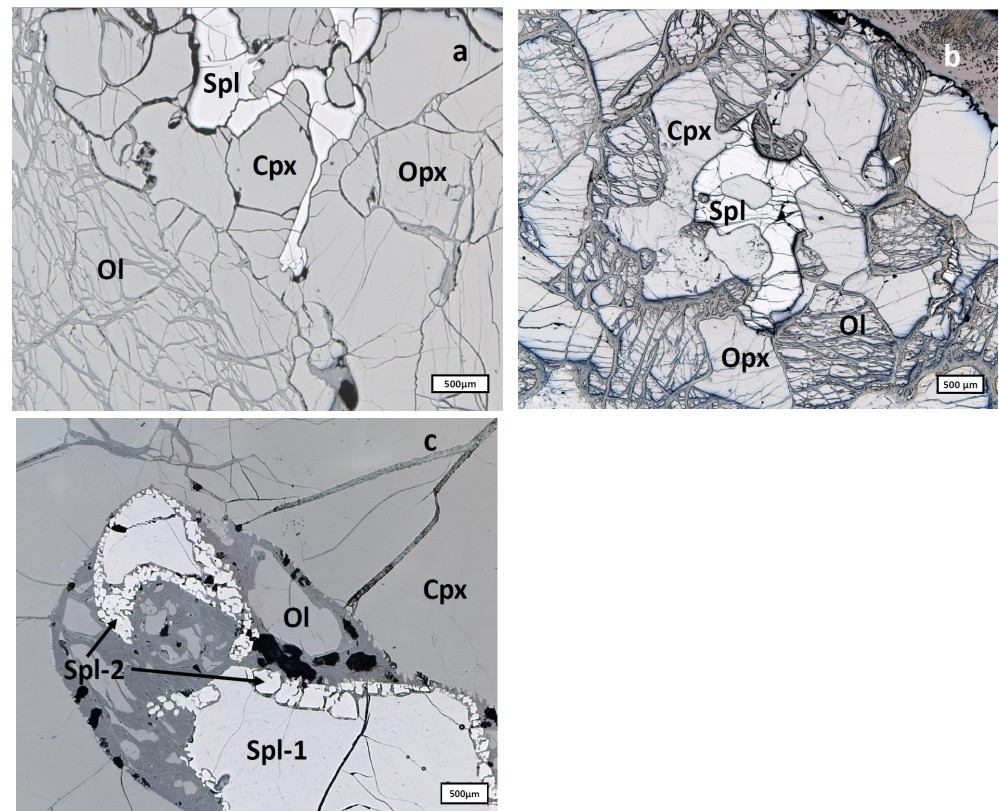

**Figure 4.** Types of spinels from Cpx-rich Spl lherzolites: (**a**), Primary spinel (Spl) represented by clean homogeneous grains of irregular shape; (**b**), Primary low-Cr spinel (Spl) from the sample XLT-5; initial stages of generation of spongy rim around clinopyroxene (Cpx) can be seen (see [4]); (**c**), Primary spinel grains (Spl-1) surrounded by the thin sporadic rim of secondary spinel (Spl-2). Ol, olivine.

The rim surrounding primary spinel grain is represented by the solid solution of the spinel component (74.7–75.3%), the chromite component (22.4–22.8%) and insignificant amounts of the magnetite and ulvöspinel components. Chemically, the rim spinels are characterized by moderate amounts of Ti (0.20–0.22 wt% $TiO_2$), high amounts of Al (45.0–45.4 wt% $Al_2O_3$) and Mg (20.6–20.7 wt% MgO), lowered relative to the host primary spinel amounts of Fe (9.5–9.6 wt% FeO) and higher Cr amounts (23.2–23.8 wt% $Cr_2O_3$) with the Cr# of 0.26.

**Table 1.** Representative major oxide (wt%) and mineral compositions of spinel in lherzolite upper mantle xenoliths from Jetty Peninsula.

| | U-3/4-2 | | | | | U-1/4-3 | | | | UN-1 | | | |
|---|---|---|---|---|---|---|---|---|---|---|---|---|---|
| | | edge | | | | rim | | rim | | edge | rim | | rim |
| | Cpx-poor Spl lherzolite | | | | | Cpx-poor Spl lherzolite | | | | Cpx-rich Spl lherzolite | | | |
| $SiO_2$ | | 0.16 | | | | 0.21 | | | | 0.15 | 0.15 | | 0.13 |
| $TiO_2$ | | | | | | 0.34 | | 0.17 | 0.25 | 0.23 | 0.22 | 0.20 | 0.28 |
| $Al_2O_3$ | 43.39 | 43.51 | 43.76 | 42.44 | 42.94 | 34.73 | 43.19 | 35.77 | 50.83 | 50.69 | 45.37 | 44.96 | 51.13 |
| $Cr_2O_3$ | 24.76 | 24.30 | 24.51 | 26.44 | 25.39 | 32.73 | 25.92 | 31.77 | 17.79 | 16.51 | 23.23 | 23.78 | 16.89 |
| FeO | 11.60 | 11.77 | 11.15 | 10.75 | 11.22 | 11.12 | 10.66 | 11.63 | 10.44 | 10.64 | 9.62 | 9.53 | 10.16 |
| MnO | 0.18 | 0.20 | 0.00 | 0.14 | | 0.20 | 0.18 | 0.15 | 0.16 | | | | 0.19 |
| MgO | 19.75 | 20.19 | 20.18 | 19.43 | 18.90 | 18.95 | 19.27 | 18.72 | 21.23 | 21.34 | 20.72 | 20.64 | 21.05 |
| NiO | | | | | | 0.40 | | | | 0.28 | | 0.38 | |
| CaO | | | | | | | | | | | | | |
| $Na_2O$ | | | | | | | | | | | | | |
| $K_2O$ | | | | | | | | 0.06 | | | | | |
| Total | 99.67 | 100.14 | 99.59 | 99.21 | 99.04 | 98.48 | 99.21 | 98.26 | 100.99 | 99.55 | 99.31 | 99.50 | 99.83 |

**Table 1.** *Cont.*

| | U-3/4-2 | | | | | U-1/4-3 | | | UN-1 | | | | |
|---|---|---|---|---|---|---|---|---|---|---|---|---|---|
| | edge | | | | | rim | | rim | edge | rim | rim | | |
| | Cpx-poor Spl lherzolite | | | | | Cpx-poor Spl lherzolite | | | Cpx-rich Spl lherzolite | | | | |
| Si | | 0.004 | | | | 0.006 | | | | 0.004 | 0.004 | | | 0.003 |
| Ti | | 0.007 | | | | | 0.004 | 0.005 | 0.004 | 0.004 | 0.004 | 0.004 | 0.004 | 0.006 |
| Al | 1.395 | 1.389 | 1.402 | 1.377 | 1.401 | 1.166 | 1.399 | 1.203 | 1.567 | 1.575 | 1.444 | 1.436 | 1.586 |
| Cr | 0.534 | 0.520 | 0.527 | 0.575 | 0.556 | 0.737 | 0.563 | 0.717 | 0.368 | 0.344 | 0.496 | 0.510 | 0.351 |
| $Fe^{3+}$ | 0.071 | 0.082 | 0.071 | 0.048 | 0.044 | 0.070 | 0.038 | 0.073 | 0.055 | 0.064 | 0.043 | 0.046 | 0.045 |
| $Fe^{2+}$ | 0.193 | 0.184 | 0.182 | 0.200 | 0.216 | 0.195 | 0.206 | 0.204 | 0.173 | 0.170 | 0.174 | 0.170 | 0.179 |
| Mn | 0.004 | 0.005 | | 0.003 | 0.004 | 0.005 | 0.004 | 0.004 | 0.004 | | | | 0.004 |
| Mg | 0.803 | 0.816 | 0.818 | 0.797 | 0.779 | 0.814 | 0.790 | 0.795 | 0.828 | 0.838 | 0.835 | 0.834 | 0.826 |
| Cr/Al | 0.468 | 0.485 | 0.455 | 0.407 | 0.442 | 0.340 | 0.411 | 0.366 | 0.586 | 0.644 | 0.414 | 0.401 | 0.602 |
| Cr# | 0.277 | 0.272 | 0.273 | 0.295 | 0.284 | 0.387 | 0.287 | 0.373 | 0.190 | 0.179 | 0.256 | 0.262 | 0.181 |
| Mg# | 0.806 | 0.816 | 0.818 | 0.799 | 0.783 | 0.807 | 0.793 | 0.796 | 0.827 | 0.831 | 0.828 | 0.831 | 0.822 |
| $F_{melt}$ | 11.16 | 10.99 | 11.02 | 11.78 | 11.42 | 14.51 | 11.52 | 14.15 | 7.40 | 6.81 | 10.36 | 10.61 | 6.92 |
| Chromite | 0.242 | 0.233 | 0.236 | 0.259 | 0.258 | 0.301 | 0.258 | 0.300 | 0.176 | 0.164 | 0.224 | 0.228 | 0.170 |
| Hercynite | | | | | | | | | | | | | |
| Magnetite | 0.032 | 0.037 | 0.032 | 0.022 | 0.020 | 0.029 | 0.017 | 0.031 | 0.026 | 0.031 | 0.019 | 0.021 | 0.022 |
| Spinel | 0.726 | 0.731 | 0.732 | 0.719 | 0.722 | 0.665 | 0.724 | 0.666 | 0.793 | 0.800 | 0.753 | 0.747 | 0.802 |
| Ulvöspinel | | | | | | 0.006 | | 0.003 | 0.005 | 0.005 | 0.004 | 0.004 | 0.006 |

| | | edge | XLT-5 | | | | | DN-1 | | | | DN-4 | |
|---|---|---|---|---|---|---|---|---|---|---|---|---|---|
| | | | Cpx-rich Spl lherz. | | | | | rim | rim | | | edge | rim |
| | | | | | | | | | Spl-Grt lherz. | | | | Spl-Grt lherz. |
| SiO$_2$ | | | | | | 0.13 | 0.13 | | | | 0.14 | | |
| TiO$_2$ | | | | | | 0.32 | 0.40 | 0.36 | 0.25 | 0.38 | 0.51 | 0.55 | 0.52 |
| Al$_2$O$_3$ | 60.17 | 59.69 | 59.52 | 60.58 | 60.57 | 50.48 | 50.80 | 44.24 | 42.51 | 51.66 | 46.20 | 45.56 | 37.75 |
| Cr$_2$O$_3$ | 8.24 | 8.12 | 8.84 | 7.92 | 8.05 | 15.49 | 15.60 | 20.30 | 22.11 | 14.88 | 19.56 | 20.01 | 28.21 |
| FeO | 11.10 | 11.45 | 10.55 | 10.95 | 10.98 | 13.18 | 12.85 | 14.83 | 14.28 | 12.51 | 13.02 | 12.79 | 13.60 |
| MnO | | 0.12 | | | | 0.13 | | 0.19 | 0.23 | 0.13 | | 0.12 | 0.13 |
| MgO | 19.68 | 19.70 | 20.37 | 19.59 | 19.69 | 19.86 | 19.84 | 18.90 | 18.61 | 20.19 | 19.62 | 20.01 | 18.39 |
| NiO | 0.46 | 0.36 | | 0.33 | 0.49 | 0.45 | 0.27 | 0.38 | 0.33 | 0.26 | 0.30 | | |
| CaO | | | | | | | | | | | | | |
| Na$_2$O | | | | | | | | | | | | | |
| K$_2$O | | | | | | | | | | | | | |
| Total | 99.65 | 99.46 | 99.27 | 99.38 | 99.79 | 100.05 | 99.90 | 99.19 | 98.32 | 100.01 | 99.35 | 99.03 | 98.61 |
| Si | | | | | | 0.003 | 0.004 | | | | 0.004 | | |
| Ti | | | | | | 0.006 | 0.008 | 0.007 | 0.005 | 0.008 | 0.010 | 0.011 | 0.011 |
| Al | 1.833 | 1.821 | 1.809 | 1.846 | 1.842 | 1.582 | 1.590 | 1.432 | 1.396 | 1.608 | 1.477 | 1.457 | 1.258 |
| Cr | 0.169 | 0.167 | 0.180 | 0.162 | 0.164 | 0.326 | 0.328 | 0.441 | 0.487 | 0.311 | 0.419 | 0.429 | 0.631 |
| $Fe^{3+}$ | | 0.012 | 0.011 | | | 0.073 | 0.059 | 0.112 | 0.106 | 0.066 | 0.075 | 0.091 | 0.089 |
| $Fe^{2+}$ | 0.240 | 0.237 | 0.217 | 0.237 | 0.237 | 0.220 | 0.226 | 0.229 | 0.227 | 0.210 | 0.220 | 0.199 | 0.233 |
| Mn | | 0.003 | | | | 0.003 | | 0.004 | 0.006 | 0.003 | | 0.003 | 0.003 |
| Mg | 0.758 | 0.760 | 0.783 | 0.755 | 0.757 | 0.787 | 0.785 | 0.775 | 0.773 | 0.794 | 0.795 | 0.810 | 0.775 |
| Cr/Al | 1.347 | 1.411 | 1.193 | 1.383 | 1.363 | 0.851 | 0.824 | 0.730 | 0.646 | 0.841 | 0.666 | 0.639 | 0.482 |
| Cr# | 0.084 | 0.084 | 0.090 | 0.081 | 0.082 | 0.171 | 0.171 | 0.235 | 0.259 | 0.162 | 0.221 | 0.227 | 0.334 |
| Mg# | 0.760 | 0.762 | 0.783 | 0.761 | 0.762 | 0.782 | 0.776 | 0.772 | 0.773 | 0.791 | 0.783 | 0.803 | 0.769 |
| $F_{melt}$ | | | | | | 6.33 | 6.34 | 9.54 | 10.48 | 5.80 | 8.90 | 9.19 | 13.04 |
| Chromite | 0.085 | 0.084 | 0.090 | 0.081 | 0.082 | 0.163 | 0.164 | 0.208 | 0.227 | 0.156 | 0.199 | 0.198 | 0.275 |
| Hercynite | 0.158 | 0.149 | 0.122 | 0.164 | 0.161 | 0.005 | 0.012 | | | 0.008 | | | |
| Magnetite | | 0.006 | 0.006 | | | 0.035 | 0.030 | 0.053 | 0.049 | 0.033 | 0.036 | 0.042 | 0.039 |
| Spinel | 0.758 | 0.762 | 0.783 | 0.755 | 0.757 | 0.789 | 0.786 | 0.732 | 0.719 | 0.796 | 0.756 | 0.749 | 0.676 |
| Ulvöspinel | | | | | | 0.006 | 0.008 | 0.007 | 0.005 | 0.008 | 0.010 | 0.010 | 0.010 |

| | DN-4 | | | DK-8/3 | | | | |
|---|---|---|---|---|---|---|---|---|
| | rim | | | | | | in kel. | in kel. |
| | Spl-Grt lherz. | | | Spl-Grt lherz. | | | | |
| SiO$_2$ | 0.16 | | | | | | | |
| TiO$_2$ | 0.49 | 0.57 | | 0.19 | 0.21 | 0.22 | | |
| Al$_2$O$_3$ | 37.50 | 45.26 | | 48.44 | 48.99 | 50.43 | 62.74 | 62.08 |
| Cr$_2$O$_3$ | 28.78 | 20.11 | | 18.92 | 18.71 | 17.13 | 6.72 | 7.13 |
| FeO | 13.18 | 13.07 | | 11.93 | 11.66 | 11.33 | 9.32 | 9.71 |
| MnO | 0.14 | 0.19 | | 0.13 | 0.13 | | 0.21 | 0.30 |
| MgO | 18.60 | 19.74 | | 19.41 | 19.66 | 20.06 | 21.05 | 20.71 |
| NiO | | 0.27 | | 0.23 | 0.00 | 0.34 | | |
| CaO | | | | | | | | |
| Na$_2$O | | | | | | | | |

**Table 1.** *Cont.*

| | DN-4 | | | DK-8/3 | | in kel. | in kel. |
|---|---|---|---|---|---|---|---|
| | rim | | | | | | |
| | Spl-Grt lherz. | | | Spl-Grt lherz. | | | |
| $K_2O$ | | | | | | | |
| Total | 98.84 | 99.21 | 99.26 | 99.36 | 99.51 | 100.03 | 99.94 |
| Si | 0.005 | | | | | | |
| Ti | 0.010 | 0.012 | 0.004 | 0.004 | 0.004 | | |
| Al | 1.247 | 1.452 | 1.540 | 1.548 | 1.585 | 1.870 | 1.859 |
| Cr | 0.642 | 0.433 | 0.404 | 0.397 | 0.361 | 0.134 | 0.143 |
| $Fe^{3+}$ | 0.082 | 0.092 | 0.049 | 0.046 | 0.045 | | |
| $Fe^{2+}$ | 0.229 | 0.206 | 0.221 | 0.215 | 0.207 | 0.197 | 0.206 |
| Mn | 0.003 | 0.004 | 0.003 | 0.003 | | 0.004 | 0.006 |
| Mg | 0.782 | 0.801 | 0.779 | 0.787 | 0.798 | 0.795 | 0.786 |
| Cr/Al | 0.458 | 0.650 | 0.631 | 0.623 | 0.661 | 1.387 | 1.363 |
| Cr# | 0.340 | 0.230 | 0.208 | 0.204 | 0.186 | 0.067 | 0.071 |
| Mg# | 0.773 | 0.795 | 0.779 | 0.785 | 0.794 | 0.801 | 0.792 |
| $F_{melt}$ | 13.21 | 9.29 | 8.29 | 8.11 | 7.15 | | |
| Chromite | 0.278 | 0.201 | 0.200 | 0.196 | 0.180 | 0.067 | 0.072 |
| Hercynite | | | | | | 0.136 | 0.139 |
| Magnetite | 0.036 | 0.043 | 0.024 | 0.023 | 0.022 | | |
| Spinel | 0.678 | 0.745 | 0.772 | 0.777 | 0.794 | 0.797 | 0.789 |
| Ulvöspinel | 0.009 | 0.011 | 0.004 | 0.004 | 0.004 | | |

Notes: Cpx, clinopyroxene; Spl, spinel; Grt, garnet; lherz., lherzolite; kel., kelyphite.

Spinels from the sample XLT-5 (Figure 4b) compositionally stay completely aside from other analyzed lherzolite spinels. These rim-free spinels are represented by the solid solution of the spinel (75.5–78.3%), chromite (8.1–9.0%) and hercynite components (12.2–16.4%) with very insignificant amounts of the magnetite component. Chemically, such spinels are characterized by the absence of Ti, very significant amounts of Al (59.5–60.6 wt% $Al_2O_3$), high amounts of Mg (19.6–20.4 wt% MgO), moderately low amounts of Fe (10.6–11.5 wt% FeO) and very low amounts of Cr (7.9–8.8 wt% $Cr_2O_3$) with the Cr# varying from 0.08 to 0.09.

3.1.3. Spinel in Spl-Grt Lherzolites

Spinels occur as sub-idiomorphic grains of 20–500 μm size scattered unevenly through the lherzolite matrix. Three types of spinel were identified here: primary spinel represented by clean homogeneous grains (Figure 5a), a thin (5–20 μm) rim of the secondary spinel (Spl-2 in Figure 5b) surrounding primary spinel grains (Spl-1 in Figure 5b), which can completely replace small spinel grains and tiny irregular spinel grains inside a kelyphite rim (Figure 5c).

Compositionally, primary spinel is represented by the solid solution of the spinel component (74.5–79.9%), the chromite component (15.0–20.1%) and insignificant amounts of the magnetite, hercynite and ulvöspinel components. Chemically, primary spinels are characterized by variable amounts of Ti (0.19–0.55 wt% $TiO_2$), high amounts of Al (45.3–51.7 wt% $Al_2O_3$) and Mg (19.4–21.1 wt% MgO) and moderately low amounts of Fe (11.4–13.2 wt% FeO) and Cr (14.9–20.1 wt% $Cr_2O_3$) with the Cr# varying from 0.16 to 0.23.

The rim at primary spinel grain is represented by the solid solution of the spinel component (66.3–75.9%), the chromite component (20.0–28.9%) and insignificant amounts of the magnetite and ulvöspinel components. Chemically, spinels of the rim are characterized by variable and are similar to the primary spinels amounts of Ti (0.25–0.64 wt% $TiO_2$), lower amounts of Al (35.2–46.5 wt% $Al_2O_3$) and Mg (17.9–19.4 wt% MgO), slightly elevated relative to the host primary spinel amounts of Fe (12.8–14.8 wt% FeO) and higher Cr amounts (19.2–29.4 wt% $Cr_2O_3$) with the Cr# varying from 0.22 to 0.36.

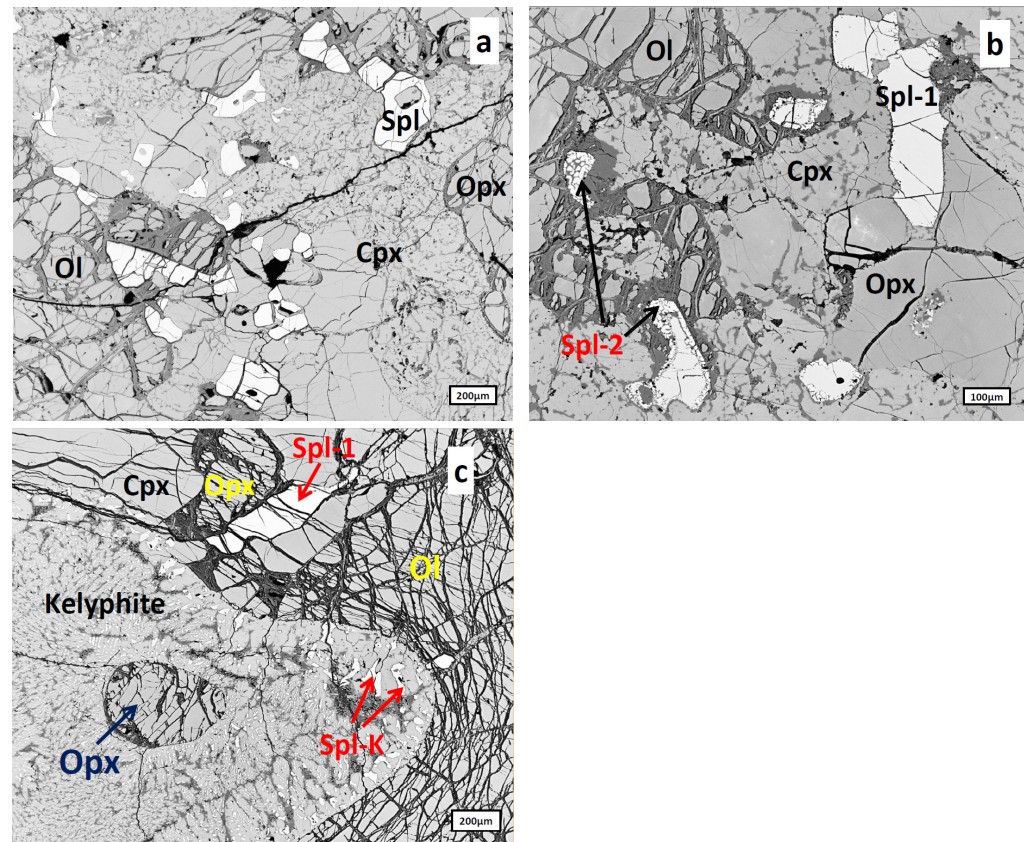

**Figure 5.** Types of spinels from Spl-Grt lherzolites: (**a**), Primary spinel (Spl) represented by clean homogeneous grains; (**b**), Primary spinel grains (Spl-1) surrounded by the thin sporadic rim of secondary spinel and completely resorbed spinel grains (Spl-2); (**c**), Irregular spinel grains (Spl-K) inside kelyphite rim replacing garnet; primary spinel (Spl-1) is also present here. A spongy rim around clinopyroxene (Cpx) can be seen in this lherzolite type (see [4]). Ol, olivine; Opx, orthopyroxene.

Spinels from the kelyphite rim are compositionally similar to the mineral observed in the matrix of Cpx-rich spinel lherzolite sample XLT-5 and are represented by the solid solution of the spinel component (78.6–79.7%), the chromite component (6.7–7.2%) and the hercynite component (13.6–14.3%). Chemically, kelyphite spinels are characterized by very high amounts of Al (62.1–62.7 wt% $Al_2O_3$) and Mg (20.7–21.1 wt% MgO), moderate amounts of Fe (9.3–9.9 wt% FeO) and very low amounts of Cr (6.7–7.1 wt% $Cr_2O_3$) with a Cr# of 0.07. All kelyphite spinels contain very low amounts of Ti which are mostly below the SEM detection limit.

### 3.1.4. Spinel in Harzburgites

Spinel in harzburgites is represented by sub-idiomorphic grains of 10–300 μm size scattered evenly through the rock. Three types of spinel were identified: primary spinel represented by clean homogeneous grains (Figure 6a), a rim of resorption (Spl-2 in Figure 6b) surrounding some primary spinel grains (Spl-1 in Figure 6b) and irregular interstitial often completely resorbed grains (Spl-2 in Figure 6c) formed because of significant development of the spinel rim).

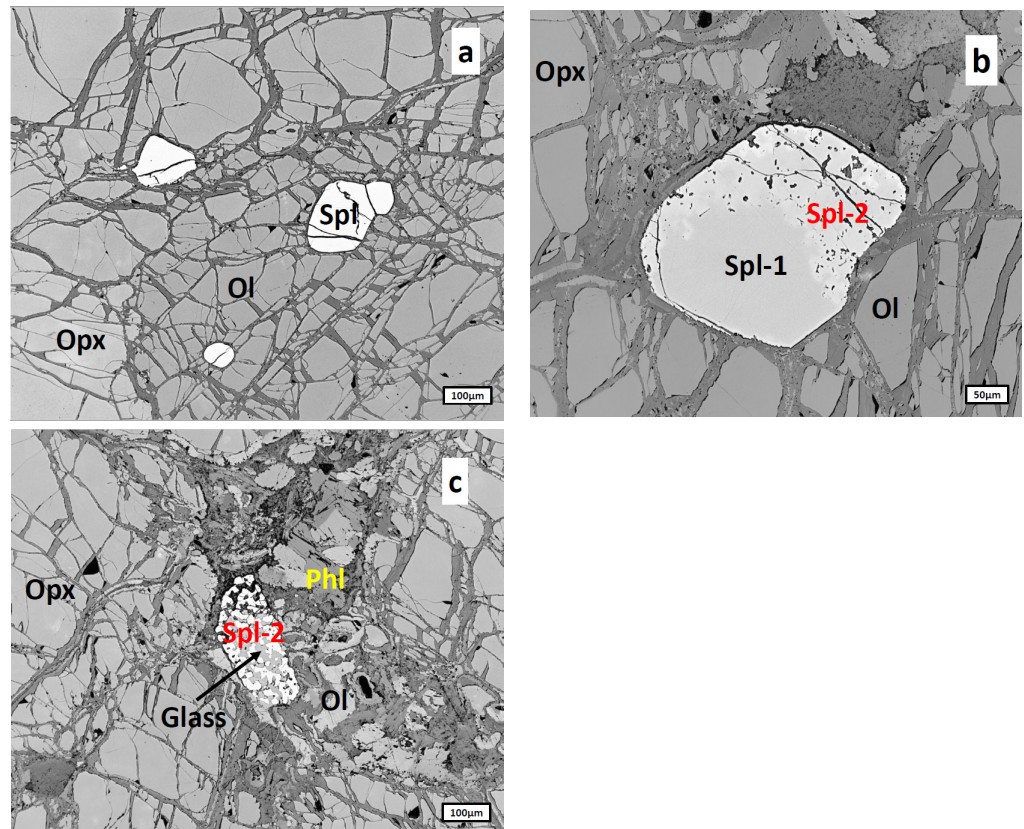

**Figure 6.** Types of spinels from harzburgites: (**a**), primary spinel (Spl) represented by clean homogeneous grains; (**b**), Primary spinel grains (Spl-1) surrounded by the spongy rim of resorption (Spl-2); (**c**), completely resorbed spinel grain (Spl-2) containing silicate glass material within the resorbed spinel matter. Ol, olivine; Opx, orthopyroxene; Phl, phlogopite.

Compositionally, primary spinel is represented by the solid solution of the spinel component (71.3–75.3%), the chromite component (17.9–20.3%) and insignificant amounts of the magnetite and ulvöspinel components. Chemically, primary spinels are characterized by the presence of high amounts of Ti (1.0–1.3 wt% $TiO_2$), Al (41.2–46.8 wt% $Al_2O_3$) and Mg (18.8–19.8 wt% MgO) and moderate amounts of Fe (15.3–16.1 wt% FeO) and Cr (17.1–21.1 wt% $Cr_2O_3$) with the Cr# of 0.20–0.26.

The rim around primary spinel grain is represented by the solid solution of the spinel component (70.1%), the chromite component (19.2%) and insignificant amounts of the magnetite and ulvöspinel components. Chemically, the rim is characterized by high amounts of Ti (1.1 wt% $TiO_2$), Al (41.0 wt% $Al_2O_3$) and Mg (18.3 wt% MgO), slightly elevated relative to the host spinel amounts of Fe (19.3 wt% FeO) and by similar amounts of Cr (19.0 wt% $Cr_2O_3$) with the Cr# of 0.24. Resorbed spinel grains are characterized by the increased presence of the chromite (23.2–27.3%) and magnetite (12.5–16.5%) components and by the decreased presence of the spinel component (54.0–56.3%) as compared to the primary spinel. Chemically, resorbed spinels are characterized by lower amounts of Al (21.6–25.2 wt% $Al_2O_3$) and Mg (14.8–15.7 wt% MgO) and increased amounts of Fe (25.8–30.1 wt% FeO) and Cr (24.2–28.6 wt% $Cr_2O_3$) with the significantly higher Cr# (0.41–0.44). Titanium is present in high amounts (2.1–3.1 wt% $TiO_2$)

3.1.5. Spinel in Dunites

Spinel here occurs as sub-idiomorphic grains of 5–250 μm size scattered unevenly through the rock. Three spinel types were identified in a sample SN-N6: primary spinel represented by clean homogeneous grains (Figure 7a), a rim of resorption (Spl-2 in Figure 7b)

surrounding some primary spinel grains (Spl-2 in Figure 7b) and irregular completely resorbed interstitial grains (Figure 7c).

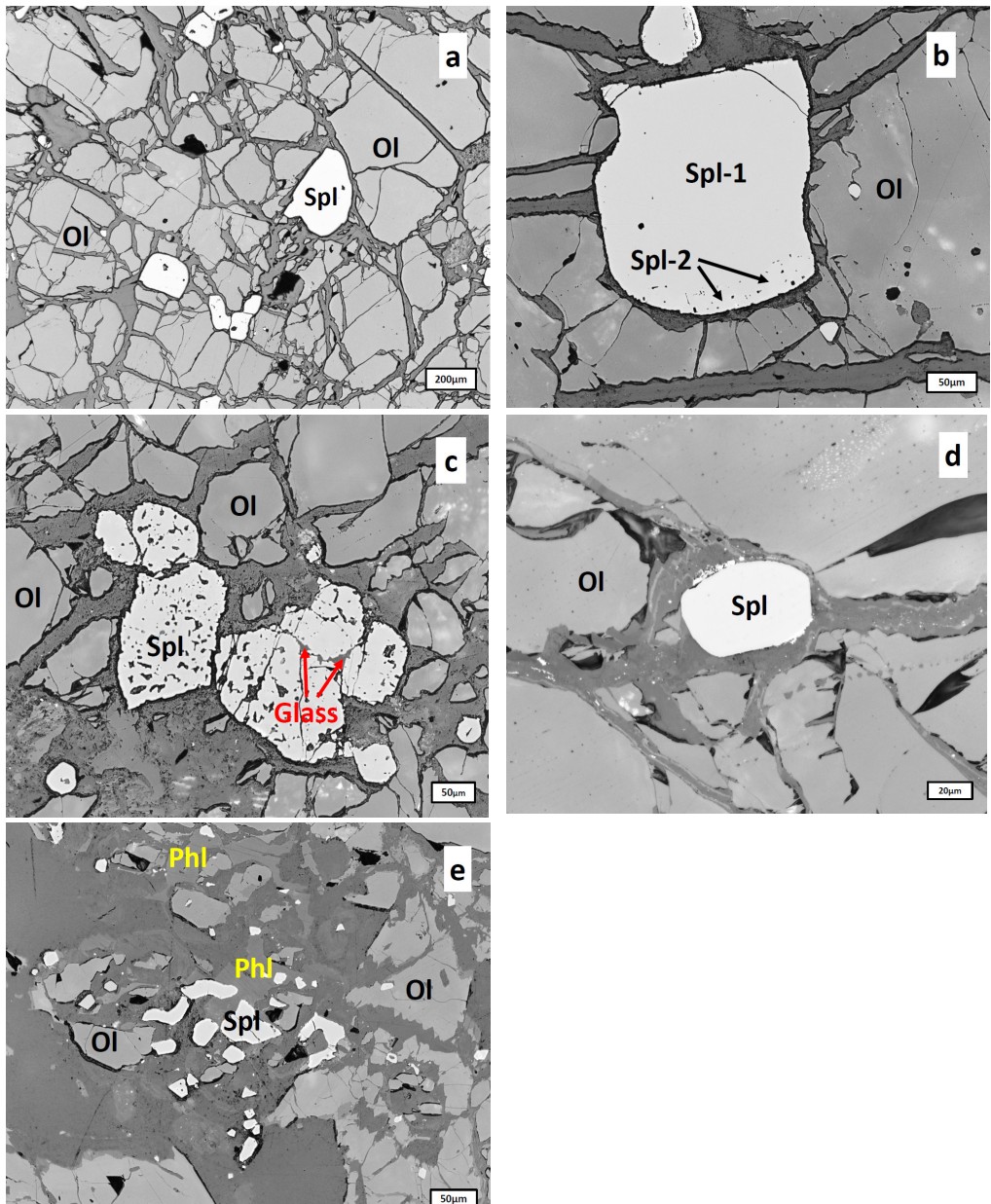

**Figure 7.** Types of spinels from dunites: (**a**), primary spinel (Spl) represented by clean homogeneous grains; (**b**), a tiny rim of resorption (Spl-2) starting to develop around the primary spinel grain (Spl-1); (**c**), completely resorbed interstitial spinel grains (Spl) containing patches of silicate glass (light-grey) inside the "spongy" material of spinel (all from the sample SN-N6); (**d**), clean homogeneous spinel grain (Spl); (**e**), irregular to skeletal spinel grains (Spl) in altered (serpentinized) silicate matrix containing phlogopite (Phl) (**d**,**e**) are from the sample SN-N9; see text for details. Ol, olivine.

**Table 2.** Representative major oxide (wt%) and mineral compositions of spinel in non-lherzolite upper mantle xenoliths from Jetty Peninsula.

| | SN-A30 | | | | | SN-N18 | | | | |
|---|---|---|---|---|---|---|---|---|---|---|
| | | | resorbed | resorbed | | | | | rim | resorbed |
| | | Harzburgite | | | | | Harzburgite | | | |
| $SiO_2$ | 0.11 | 0.13 | 0.11 | 0.08 | 0.08 | 0.10 | 0.10 | 0.13 | 0.11 | 0.02 |
| $TiO_2$ | 1.18 | 1.13 | 1.10 | 3.08 | 2.84 | 0.96 | 1.00 | 0.92 | 1.05 | 2.21 |
| $Al_2O_3$ | 42.16 | 41.22 | 41.56 | 21.59 | 23.23 | 45.50 | 45.44 | 46.80 | 41.02 | 25.21 |
| $Cr_2O_3$ | 20.62 | 21.10 | 20.44 | 25.49 | 24.19 | 17.92 | 17.98 | 17.09 | 18.99 | 28.19 |
| $FeO$ | 15.96 | 15.28 | 16.13 | 29.62 | 30.13 | 15.54 | 15.97 | 15.34 | 19.29 | 25.82 |
| $MnO$ | 0.27 | 0.14 | 0.21 | 0.12 | 0.10 | 0.18 | 0.22 | 0.28 | 0.10 | 0.20 |
| $MgO$ | 19.26 | 19.75 | 19.52 | 14.99 | 15.19 | 19.74 | 19.58 | 18.84 | 18.30 | 15.51 |
| $NiO$ | 0.22 | 0.14 | 0.15 | 0.25 | 0.28 | 0.29 | 0.21 | 0.36 | 0.39 | 0.39 |
| $CaO$ | 0.02 | | | | 0.07 | | | 0.01 | 0.02 | 0.08 |
| $Na_2O$ | 0.03 | 0.03 | 0.03 | 0.06 | 0.05 | | | | 0.03 | 0.01 |
| $K_2O$ | | | | 0.07 | | 0.01 | 0.01 | | 0.02 | |
| Total | 99.82 | 98.93 | 99.25 | 95.35 | 96.16 | 100.24 | 100.50 | 99.77 | 99.32 | 97.64 |
| Si | 0.003 | 0.004 | 0.003 | 0.002 | 0.003 | 0.003 | 0.003 | 0.004 | 0.003 | 0.001 |
| Ti | 0.024 | 0.023 | 0.023 | 0.073 | 0.066 | 0.019 | 0.020 | 0.019 | 0.022 | 0.051 |
| Al | 1.364 | 1.342 | 1.350 | 0.805 | 0.849 | 1.447 | 1.443 | 1.495 | 1.346 | 0.904 |
| Cr | 0.448 | 0.461 | 0.445 | 0.637 | 0.593 | 0.382 | 0.383 | 0.366 | 0.418 | 0.678 |
| $Fe^{3+}$ | 0.133 | 0.143 | 0.153 | 0.407 | 0.419 | 0.127 | 0.128 | 0.094 | 0.186 | 0.315 |
| $Fe^{2+}$ | 0.234 | 0.210 | 0.219 | 0.376 | 0.364 | 0.224 | 0.232 | 0.254 | 0.264 | 0.342 |
| Mn | 0.006 | 0.003 | 0.005 | 0.003 | 0.003 | 0.004 | 0.005 | 0.006 | 0.002 | 0.005 |
| Mg | 0.788 | 0.814 | 0.802 | 0.697 | 0.703 | 0.794 | 0.786 | 0.762 | 0.759 | 0.704 |
| Cr/Al | 0.77 | 0.72 | 0.79 | 1.16 | 1.25 | 0.87 | 0.89 | 0.90 | 1.02 | 0.92 |
| Cr# | 0.25 | 0.26 | 0.25 | 0.44 | 0.41 | 0.21 | 0.21 | 0.20 | 0.24 | 0.43 |
| Mg# | 0.77 | 0.80 | 0.79 | 0.65 | 0.66 | 0.78 | 0.77 | 0.75 | 0.74 | 0.67 |
| $F_{melt}$ | 10.0 | 10.4 | 10.1 | 15.8 | 15.1 | 8.3 | 8.4 | 7.7 | 9.6 | 15.5 |
| Chromite | 0.203 | 0.202 | 0.199 | 0.246 | 0.232 | 0.179 | 0.180 | 0.181 | 0.192 | 0.272 |
| Hercynite | | | | | | | | | | |
| Magnetite | 0.061 | 0.064 | 0.068 | 0.158 | 0.165 | 0.060 | 0.060 | 0.046 | 0.087 | 0.125 |
| Spinel | 0.715 | 0.713 | 0.713 | 0.540 | 0.551 | 0.743 | 0.741 | 0.753 | 0.701 | 0.563 |
| Ulvöspinel | 0.022 | 0.021 | 0.020 | 0.056 | 0.052 | 0.019 | 0.019 | 0.019 | 0.020 | 0.041 |

| | SN-N6 | | | | | SN-N9 | | |
|---|---|---|---|---|---|---|---|---|
| | | rim | resorbed | resorbed | | secondary | | secondary |
| | | Dunite | | | | Dunite | | |
| $SiO_2$ | 0.11 | 0.07 | 0.06 | 0.13 | 0.07 | 0.06 | 0.03 | 0.12 |
| $TiO_2$ | 1.21 | 1.18 | 1.24 | 1.40 | 1.71 | 1.45 | 1.10 | 1.37 |
| $Al_2O_3$ | 28.66 | 30.16 | 27.44 | 25.85 | 23.32 | 21.05 | 15.61 | 16.98 |
| $Cr_2O_3$ | 35.36 | 33.90 | 32.27 | 35.70 | 36.97 | 37.23 | 45.71 | 41.13 |
| $FeO$ | 16.77 | 16.48 | 18.11 | 20.02 | 21.03 | 23.78 | 24.68 | 28.30 |
| $MnO$ | 0.10 | 0.15 | 0.19 | 0.20 | 0.20 | 0.27 | 0.43 | 0.46 |
| $MgO$ | 17.32 | 17.43 | 17.22 | 16.50 | 16.19 | 14.96 | 11.62 | 10.60 |
| $NiO$ | 0.31 | 0.11 | 0.20 | 0.26 | 0.32 | 0.29 | 0.23 | 0.34 |
| $CaO$ | | 0.04 | 0.02 | 0.01 | | 0.03 | | 0.09 |
| $Na_2O$ | | | 0.02 | | | 0.03 | 0.01 | |
| $K_2O$ | | 0.02 | 0.02 | 0.02 | 0.01 | 0.03 | 0.04 | 0.01 |
| Total | 99.84 | 99.52 | 96.80 | 100.10 | 99.82 | 99.18 | 99.47 | 99.41 |
| Si | 0.003 | 0.002 | 0.002 | 0.004 | 0.002 | 0.002 | 0.001 | 0.004 |
| Ti | 0.027 | 0.026 | 0.028 | 0.031 | 0.039 | 0.033 | 0.027 | 0.033 |
| Al | 0.987 | 1.033 | 0.971 | 0.902 | 0.825 | 0.761 | 0.589 | 0.642 |
| Cr | 0.817 | 0.779 | 0.767 | 0.835 | 0.877 | 0.903 | 1.157 | 1.043 |
| $Fe^{3+}$ | 0.137 | 0.132 | 0.202 | 0.193 | 0.217 | 0.266 | 0.199 | 0.241 |
| $Fe^{2+}$ | 0.273 | 0.269 | 0.254 | 0.302 | 0.311 | 0.344 | 0.462 | 0.517 |
| Mn | 0.002 | 0.004 | 0.005 | 0.005 | 0.005 | 0.007 | 0.011 | 0.013 |
| Mg | 0.754 | 0.755 | 0.771 | 0.728 | 0.724 | 0.684 | 0.554 | 0.507 |
| Cr/Al | 0.47 | 0.49 | 0.56 | 0.56 | 0.57 | 0.64 | 0.54 | 0.69 |

**Table 2.** *Cont.*

| | | | SN-N6 rim Dunite | resorbed | resorbed | | SN-N9 secondary Dunite | secondary |
|---|---|---|---|---|---|---|---|---|
| Cr# | 0.45 | 0.43 | 0.44 | 0.48 | 0.52 | 0.54 | 0.66 | 0.62 |
| Mg# | 0.73 | 0.74 | 0.75 | 0.71 | 0.70 | 0.67 | 0.55 | 0.50 |
| $F_{melt}$ | 16.1 | 15.6 | 15.8 | 16.7 | 17.4 | 17.9 | 19.9 | 19.2 |
| Chromite | 0.324 | 0.315 | 0.299 | 0.328 | 0.335 | 0.347 | 0.460 | 0.441 |
| Hercynite | | | | | | | | |
| Magnetite | 0.055 | 0.053 | 0.079 | 0.077 | 0.082 | 0.101 | 0.077 | 0.101 |
| Spinel | 0.599 | 0.610 | 0.601 | 0.571 | 0.553 | 0.526 | 0.441 | 0.430 |
| Ulvöspinel | 0.021 | 0.021 | 0.022 | 0.024 | 0.030 | 0.026 | 0.021 | 0.028 |

Compositionally, primary spinel is represented by the solid solution of the spinel component (43.0–60.2%), the chromite component (31.5–46.0%), lower amounts of the magnetite component (5.5–10.1%) and insignificant amounts of the ulvöspinel component. Chemically, primary spinels are characterized by high amounts of Ti (1.2–1.3 wt% $TiO_2$), not very high amounts of Al (15.6–30.2 wt% $Al_2O_3$) and Mg (10.6–17.4 wt% MgO), moderate amounts of Fe (16.5–28.3 wt% FeO) and high amounts of Cr (33.9–45.7 wt% $Cr_2O_3$) with the Cr# of 0.43–0.45.

The rim of resorption is represented by the solid solution of the spinel component (60.1%), the chromite component (29.9%) and insignificant amounts of the magnetite and ulvöspinel components. Chemically, the rim is characterized by the high amount of Ti (1.2 wt% $TiO_2$), moderately low amounts of Al (27.4 wt% $Al_2O_3$) and Mg (17.2 wt% MgO), slightly elevated relative to the host primary spinel amounts of Fe (18.1 wt% FeO) and similar to the primary spinel Cr amount (32.3 wt% $Cr_2O_3$) with the Cr# of 0.44. Resorbed spinel grains are characterized by the high chromite component (32.8–33.5%) and by the slightly decreased spinel component (55.3–57.1%). The presence of the magnetite and ulvöspinel components is insignificant. Chemically, resorbed spinels are characterized by lowered amounts of Al (23.3–25.9 wt% $Al_2O_3$) and Mg (16.2–16.5 wt% MgO) and slightly elevated amounts of Fe (20.1–21.4 wt% FeO) and Cr (35.7–37.0 wt% $Cr_2O_3$) with a Cr# of 0.48–0.52. Amounts of Ti are somewhat higher than in the primary spinels (1.4–1.7 wt% $TiO_2$).

Dunite sample SN-N9 contains spinel grains of the two types. Spinel of the first type (Figure 7d) is texturally similar to the primary spinel in dunite sample SN-N6. Compositionally, spinel of the first type is represented by the solid solution of the spinel component (52.6%), lower amount of the chromite component (34.7%), lower but overall elevated amount of the magnetite component (10.1%) and an insignificant amount of the ulvöspinel component. Chemically, spinel is characterized by high amount of Ti (1.5 wt% $TiO_2$), not very high amounts of Al (21.1 wt% $Al_2O_3$) and Mg (15.0 wt% MgO), moderate amount of Fe (23.8 wt% FeO) and high amount of Cr (37.2 wt% $Cr_2O_3$) with the Cr# of 0.54. Spinel of the second type is represented by irregular to skeletal grains in altered (serpentinized) silicate matrix (Figure 7e). Compositionally, such spinel is represented by the solid solution of similar amounts of the spinel (43.0–44.1%) and chromite (44.1–46.0%) components, lower but overall elevated amounts of the magnetite component (7.7–10.1%) and insignificant amounts of the ulvöspinel component. Chemically, spinel of the second type is characterized by the high amounts of Ti (1.1–1.4 wt% $TiO_2$), low amounts of Al (15.6–17.0 wt% $Al_2O_3$) and Mg (10.6–11.6 wt% MgO), slightly elevated amounts of Fe (24.7–28.3 wt% FeO) and very high amounts of Cr (41.1–45.7 wt% $Cr_2O_3$) with the Cr# varying from 0.62 to 0.66.

*3.2. Trace Element Characteristics of Spinels*

Most trace elements analyzed are present in spinels in very low amounts. Only Ti (which can behave either as a major or as a trace element), V, Mn, Co, Ni, Zn and Ga display concentrations in the range of tens to hundreds (up to thousands) ppm. Other trace elements concentrations vary from below the detection limit (bdl) to <10 parts per million

(ppm). Overall, very low concentrations of the REE (the elements which are otherwise very informative in terms of petrological implication) are typical for all studied spinels. In this, the currently studied xenolith spinel is different from the mineral reported in [31]. However, those authors provided information on the upper mantle beneath the Western Antarctica which is structurally and tectonically significantly different from the East Antarctica. Results of the LA-ICP-MS analyses of spinels are summarized in Tables 3 and 4 and in the Electronic Supplementary Table S2.

**Table 3.** Representative trace element compositions of spinel in lherzolite upper mantle xenoliths from Jetty Peninsula (ppm).

| | U-3/4-2 | edge | | | | U-1/4-3 resorbed | resorbed | |
|---|---|---|---|---|---|---|---|---|
| | Cpx-poor Spl lherzolite | | | | | Cpx-poor Spl lherzolite | | |
| Sc | 1.17 | 1.41 | 1.41 | 1.08 | 1.26 | | 1.34 | 0.36 |
| Ti | 186 | 180 | 175 | 217 | 209 | 1815 | 209 | 1712 |
| V | 684 | 707 | 674 | 755 | 736 | 924 | 748 | 893 |
| Mn | 739 | 745 | 693 | 831 | 771 | 828 | 787 | 889 |
| Co | 225 | 227 | 218 | 240 | 228 | 191 | 230 | 208 |
| Ni | 2061 | 2071 | 2016 | 2007 | 2005 | 1456 | 1988 | 1556 |
| Cu | 5.24 | 5.47 | 5.71 | 3.93 | 4.68 | 4.30 | 4.35 | 4.68 |
| Zn | 642 | 674 | 641 | 691 | 675 | 679 | 687 | 693 |
| Ga | 41.8 | 43.8 | 41.9 | 42.8 | 42.3 | 44.0 | 41.7 | 39.8 |
| Se | | | 0.64 | | | | 1.21 | |
| Rb | | 0.038 | | 0.06 | 0.06 | | 0.07 | 0.02 |
| Sr | | | 0.015 | 0.01 | 0.01 | 0.11 | 0.01 | 0.06 |
| Y | 0.005 | | 0.003 | | | | 0.002 | |
| Nb | 0.35 | 0.38 | 0.39 | 0.23 | 0.19 | 0.36 | 0.22 | 0.22 |
| Zr | 0.18 | 0.15 | 0.16 | 0.16 | 0.16 | | 0.21 | 0.02 |
| Ag | | | | | | | | |
| Cd | 0.03 | 0.10 | 0.02 | | | | 0.09 | 0.01 |
| In | 0.004 | | | 0.01 | | 0.02 | 0.01 | |
| Sn | 0.17 | 0.22 | 0.23 | 0.25 | 0.20 | | 0.15 | 0.03 |
| Sb | | | | | | | | |
| Te | | | | | | | | |
| Ba | | | | | | 0.01 | | 0.05 |
| La | | | | | | | | 0.05 |
| Ce | | | 0.001 | | | | 0.001 | 0.02 |
| Pr | | | 0.001 | 0.003 | 0.001 | 0.05 | | 0.16 |
| Nd | 0.01 | | | | 0.02 | | 0.01 | 0.01 |
| Sm | 0.01 | | | 0.010 | | | | |
| Eu | | 0.01 | | | | 0.12 | 0.004 | |
| Gd | | 0.01 | | | 0.02 | | | |
| Tb | 0.004 | 0.002 | | | | | | 0.01 |
| Dy | 0.003 | | | | 0.01 | | 0.01 | |
| Ho | | | | | | | | |
| Er | | | | 0.01 | 0.01 | | | |
| Tm | 0.01 | | | | | | | |
| Yb | 0.004 | | | | | | | |
| Lu | 0.001 | | 0.003 | 0.001 | | | 0.001 | |
| Hf | 0.01 | 0.02 | 0.01 | | | 0.01 | 0.01 | 0.01 |
| Ta | 0.01 | 0.02 | 0.02 | | 0.003 | | 0.01 | |
| Pt | | | | 0.01 | | | | |
| Au | | 0.003 | | | 0.01 | | | |
| Tl | | | | | | | | |
| Pb | | | | | | | | 0.03 |
| Bi | | | | | | | | |
| Ni/Co | 9.14 | 9.14 | 9.24 | 8.36 | 8.79 | 7.63 | 8.66 | 7.47 |

**Table 3.** *Cont*.

| | | edge | UN-1 resorbed Cpx-rich Spl lherzolite | resorbed | | | edge | XLT-5 Cpx-rich Spl lherzolite | | |
|---|---|---|---|---|---|---|---|---|---|---|
| Sc | 0.73 | 0.91 | 2.28 | 2.10 | 0.73 | | 0.30 | | | 0.11 |
| Ti | 1264 | 1351 | 1054 | 963.2 | 1279 | 182 | 180 | 216 | 195 | 172 |
| V | 489 | 530 | 699 | 630 | 498 | 349 | 397 | 351 | 349 | 364 |
| Mn | 618 | 661 | 649 | 610 | 621 | 647 | 694 | 639 | 644 | 647 |
| Co | 204 | 219 | 214 | 169 | 208 | 435 | 470 | 458 | 450 | 463 |
| Ni | 2539 | 2721 | 2131 | 1686 | 2595 | 2927 | 3188 | 2968 | 2920 | 3025 |
| Cu | 3.32 | 3.16 | 3.27 | 3.33 | 2.79 | 0.54 | 0.69 | 0.59 | 0.81 | 0.23 |
| Zn | 538 | 609 | 600 | 499 | 587 | 1410 | 1582 | 1505 | 1587 | 1637 |
| Ga | 66.4 | 71.4 | 69.5 | 60.2 | 69.2 | 72.3 | 79.1 | 69.1 | 70.7 | 74.2 |
| Se | | | | | | | | | | |
| Rb | | 0.13 | | | | | | 0.03 | | 0.05 |
| Sr | | 0.12 | 0.13 | 0.08 | | | | | | |
| Y | 0.004 | | 0.04 | | 0.01 | 0.01 | | 0.004 | | |
| Nb | 0.15 | 0.14 | | 0.01 | 0.11 | 0.04 | 0.02 | 0.03 | 0.05 | 0.04 |
| Zr | 0.09 | 0.18 | | 0.003 | 0.14 | 0.10 | 0.04 | 0.06 | | |
| Ag | | | | | | | | | | |
| Cd | 0.05 | | | 0.22 | | 0.04 | | 0.07 | | |
| In | 0.01 | | | 0.06 | 0.01 | | 0.02 | | | |
| Sn | 0.19 | 0.25 | 0.61 | | 0.30 | | 0.31 | 0.16 | | 0.14 |
| Sb | 0.11 | | | | | | | | | |
| Te | | | | | 0.39 | | 0.65 | | 0.25 | |
| Ba | | 0.10 | 0.09 | | 0.02 | | | | | |
| La | | | | 0.01 | | | 0.001 | 0.001 | | |
| Ce | 0.001 | | | 0.03 | | 0.002 | | | | |
| Pr | | 0.001 | 0.01 | | | 0.002 | | 0.001 | | |
| Nd | 0.01 | | | | | | | | | 0.01 |
| Sm | | | 0.067 | | | | 0.01 | | 0.01 | 0.01 |
| Eu | | | | | 0.003 | 0.002 | | | | |
| Gd | | | | | | | | | | 0.01 |
| Tb | | | 0.02 | 0.03 | | | | | | 0.001 |
| Dy | | 0.01 | 0.04 | | | | 0.004 | | | |
| Ho | | | | | 0.001 | | | | | |
| Er | | 0.003 | 0.03 | 0.03 | | | | 0.003 | | 0.003 |
| Tm | | 0.001 | | | 0.001 | 0.001 | | | | |
| Yb | | | 0.05 | 0.04 | | | | 0.01 | | 0.01 |
| Lu | 0.002 | | | | | | | | 0.002 | |
| Hf | 0.004 | | | | 0.004 | | | | | 0.004 |
| Ta | 0.01 | 0.01 | | | | | | | | |
| Pt | | | | | | | 0.01 | | | |
| Au | | 0.003 | | | | | 0.01 | | | |
| Tl | | | | | | | | | | |
| Pb | | | | 0.09 | | | | 0.02 | | 0.02 |
| Bi | 0.02 | | | | | | | | | |
| Ni/Co | 12.47 | 12.40 | 9.97 | 9.99 | 12.49 | 6.73 | 6.79 | 6.48 | 6.49 | 6.53 |

| | | | DN-1 Spl-Grt lherzolite | resorbed | resorbed | | edge | DN-4 resorbed | Spl-Grt lherzolite | resorbed |
|---|---|---|---|---|---|---|---|---|---|---|
| Sc | 1.48 | 1.20 | 1.00 | | | 0.77 | 0.85 | | 0.91 | 9.85 |
| Ti | 1989 | 1897 | 1992 | 1654 | 1575 | 2745 | 2850 | 1549 | 2899 | 6440 |
| V | 562 | 553 | 581 | 751 | 616 | 695 | 730 | 1216 | 742 | 1260 |

**Table 3.** *Cont*.

| | DN-1 Spl-Grt lherzolite | | | resorbed | resorbed | | edge | DN-4 resorbed | Spl-Grt lherzolite | resorbed |
|---|---|---|---|---|---|---|---|---|---|---|
| Mn | 809 | 705 | 696 | 1052 | 880.1 | 689 | 700 | 973 | 696 | 1001 |
| Co | 217 | 221 | 222 | 188 | 186.4 | 209 | 219 | 188 | 213 | 186 |
| Ni | 2796 | 2813 | 2843 | 2188 | 2034 | 2460 | 2565 | 1557 | 2527 | 1774 |
| Cu | 1.75 | 2.05 | 2.40 | bdl | bdl | 5.72 | 5.70 | | 5.40 | bdl |
| Zn | 687 | 697 | 708 | 736 | 646 | 665 | 704 | 696 | 690 | 928 |
| Ga | 91.1 | 88.6 | 88.7 | 74.3 | 80.0 | 105 | 112 | 105 | 108 | 96.2 |
| Se | | | 1.20 | | | | 0.49 | | | |
| Rb | | 0.05 | 0.04 | | | 0.10 | 0.04 | 3.42 | 0.13 | 6.15 |
| Sr | 0.01 | 0.004 | | 28.7 | | 0.01 | | 2.94 | 0.01 | 33.3 |
| Y | | | 0.01 | | | 0.002 | 0.002 | 0.62 | 0.01 | 3.27 |
| Nb | 0.12 | 0.12 | 0.13 | 0.41 | 0.37 | 0.18 | 0.20 | 0.79 | 0.20 | 6.29 |
| Zr | 0.20 | 0.19 | 0.22 | | | 0.28 | 0.30 | | 0.21 | 27.9 |
| Ag | | | | | | | | | | |
| Cd | 0.02 | 0.01 | | | | 0.05 | | | 0.02 | |
| In | 0.01 | 0.02 | 0.02 | | 0.27 | | 0.01 | | 0.01 | |
| Sn | 0.26 | 0.29 | 0.34 | | | 0.36 | 0.26 | | 0.40 | |
| Sb | | | | | | | | | | |
| Te | | | | | | | | 0.25 | | |
| Ba | 0.02 | | 0.001 | 9.55 | | 0.008 | | 9.90 | 0.02 | 33.2 |
| La | | | 0.001 | | | 0.001 | | 0.14 | | 2.64 |
| Ce | | 0.001 | | 1.45 | | 0.001 | 0.001 | 0.82 | | 6.50 |
| Pr | 0.01 | 0.001 | | | | | 0.01 | 0.06 | | 0.76 |
| Nd | 0.01 | | | | 0.37 | | 0.01 | 0.67 | | 3.55 |
| Sm | | | 0.001 | | 0.91 | | | 0.44 | | |
| Eu | 0.01 | | 0.01 | | | | | | | 0.55 |
| Gd | 0.01 | | 0.01 | | | 0.002 | | | 0.001 | |
| Tb | | | | | | 0.003 | | | 0.001 | |
| Dy | | | | | 0.63 | 0.004 | | | | 0.38 |
| Ho | | | | 0.11 | | | 0.004 | | 0.004 | |
| Er | | | 0.001 | 1.09 | 0.26 | 0.001 | 0.004 | | 0.002 | |
| Tm | | | 0.001 | | | 0.001 | | | 0.001 | |
| Yb | | | | | | 0.001 | | | | 0.86 |
| Lu | 0.003 | 0.003 | 0.003 | 0.17 | | 0.001 | 0.01 | | | 0.20 |
| Hf | 0.01 | 0.01 | 0.01 | 0.44 | | 0.01 | 0.01 | 0.55 | 0.02 | 1.45 |
| Ta | 0.01 | 0.01 | 0.01 | | 0.07 | 0.01 | 0.01 | 0.19 | 0.02 | 0.50 |
| Pt | | | | | | 0.01 | 0.01 | | | |
| Au | | | | | | | | 0.004 | 0.004 | |
| Tl | | | | | | | | | | |
| Pb | | | | 0.65 | | | | | | 0.51 |
| Bi | | | | | | | | | | |
| Ni/Co | 12.85 | 12.73 | 12.80 | 11.66 | 10.91 | 11.76 | 11.71 | 8.28 | 11.86 | 9.52 |

| | DK-8/3 Spl-Grt lherzolite | | | in kelyph. | In kelyph. |
|---|---|---|---|---|---|
| Sc | 0.92 | 0.73 | 0.80 | 2.60 | 3.90 |
| Ti | 1163 | 1264 | 1044 | 156 | 138 |
| V | 646 | 674 | 600 | 196 | 200 |
| Mn | 717 | 701 | 688 | 1412 | 1692 |
| Co | 227 | 225 | 213 | 116 | 116 |
| Ni | 2483 | 2657 | 2437 | 223 | 228 |

**Table 3.** *Cont.*

| | | | DK-8/3 | | |
| --- | --- | --- | --- | --- | --- |
| | | | | in kelyph. | In kelyph. |
| | | | Spl-Grt lherzo-lite | | |
| Cu | 4.56 | 4.96 | 4.16 | 0.67 | 1.53 |
| Zn | 609 | 604 | 593 | 50.9 | 53.0 |
| Ga | 70.7 | 72.4 | 69.5 | 9.99 | 10.5 |
| Se | | | | | |
| Rb | 0.07 | | 0.05 | | |
| Sr | | | 0.02 | | 0.04 |
| Y | | | | | 0.02 |
| Nb | 0.16 | 0.20 | 0.15 | 0.06 | 0.05 |
| Zr | 0.20 | 0.23 | 0.12 | 0.12 | |
| Ag | | | | 0.03 | |
| Cd | 0.08 | | 0.02 | | |
| In | 0.02 | 0.05 | | | 0.02 |
| Sn | 0.21 | | 0.26 | 0.41 | |
| Sb | | | 0.13 | | |
| Te | 0.35 | | 0.35 | | |
| Ba | | | 0.02 | 0.09 | 0.05 |
| La | | 0.01 | | 0.01 | 0.01 |
| Ce | | | | 0.01 | 0.01 |
| Pr | | 0.01 | 0.002 | | |
| Nd | 0.01 | | | 0.03 | |
| Sm | 0.01 | | 0.01 | | |
| Eu | | 0.01 | 0.002 | | |
| Gd | | | | | |
| Tb | | | 0.004 | 0.01 | 0.01 |
| Dy | 0.01 | 0.04 | 0.004 | 0.02 | |
| Ho | | | | | |
| Er | | | | | 0.04 |
| Tm | | 0.01 | 0.002 | | |
| Yb | 0.01 | | | 0.02 | 0.03 |
| Lu | 0.002 | 0.01 | | | |
| Hf | 0.01 | 0.02 | | | |
| Ta | | 0.01 | | 0.01 | |
| Pt | | | 0.01 | | |
| Au | | | | | |
| Tl | | | | | |
| Pb | | 0.04 | | | |
| Bi | | | | | |
| Ni/Co | 10.94 | 11.82 | 11.43 | 1.92 | 1.97 |

Note: Cpx, clinopyroxene; Spl, spinel; Grt, garnet; kelyph., kelyphite.

**Table 4.** Representative trace element compositions of spinel in non-lherzolite upper mantle xenoliths from Jetty Peninsula (ppm).

| | SN-A30 | | | | | SN-N18 | | | |
| --- | --- | --- | --- | --- | --- | --- | --- | --- | --- |
| | | | | resorbed | | edge | | rim | resorbed |
| | | Harzburgite | | | | | Harzburgite | | |
| Sc | 1.04 | 1.36 | 1.07 | 3.50 | 1.20 | 1.20 | 0.76 | 2.99 | 6.50 |
| Ti | 6175 | 6188 | 6544 | 24836 | 5061 | 5206 | 5106 | 9251 | 17488 |
| V | 604 | 609 | 660 | 904 | 614 | 635 | 588 | 652 | 952 |
| Mn | 825 | 785 | 828 | 1932 | 790 | 785 | 752 | 1066 | 1631 |
| Co | 210 | 206 | 222 | 237 | 203 | 212 | 197 | 221 | 222 |
| Ni | 1572 | 1585 | 1688 | 1690 | 1588 | 1686 | 1627 | 1956 | 1931 |

**Table 4.** *Cont.*

| | SN-A30 | | | | | SN-N18 | | | |
|---|---|---|---|---|---|---|---|---|---|
| | | | | resorbed | | edge | | rim | resorbed |
| | Harzburgite | | | | | Harzburgite | | | |
| Cu | 2.03 | 2.49 | 2.35 | 2.36 | 2.33 | 2.41 | 2.65 | 5.27 | 6.05 |
| Zn | 731 | 717 | 763 | 772 | 742 | 750 | 727 | 847 | 878 |
| Ga | 124 | 124 | 135 | 101 | 143 | 140 | 137 | 131 | 117 |
| Se | 2.74 | 2.35 | 1.95 | 2.45 | 2.36 | 2.54 | 3.31 | 3.08 | 15.9 |
| Rb | 0.09 | 0.11 | 0.09 | 4.25 | 0.09 | 0.11 | 0.10 | 3.31 | 0.78 |
| Sr | 0.02 | 0.02 | 0.02 | 127 | 0.02 | 0.01 | 0.01 | 17.0 | 9.70 |
| Y | 0.02 | | | 2.17 | | 0.03 | 0.02 | 0.37 | 0.17 |
| Nb | 0.19 | 0.21 | 0.23 | 37.1 | 0.17 | 0.15 | 0.13 | 3.40 | 0.16 |
| Zr | 0.47 | 0.53 | 0.52 | 15.0 | 0.44 | 0.41 | 0.21 | 4.14 | 2.91 |
| Ag | 0.05 | 0.05 | 0.03 | 0.18 | 0.08 | 0.03 | 0.05 | 0.07 | 0.85 |
| Cd | | 0.10 | | | 0.20 | 0.25 | | 0.38 | |
| In | 0.03 | 0.04 | 0.03 | 0.12 | 0.02 | 0.01 | 0.02 | 0.03 | 0.05 |
| Sn | 0.77 | 0.71 | 0.49 | 2.15 | 0.26 | 0.65 | 0.35 | 0.45 | 4.97 |
| Sb | 0.21 | 0.19 | 0.22 | 0.91 | 0.25 | 0.25 | 0.22 | 0.38 | 1.52 |
| Te | | | | 1.12 | 0.54 | 1.11 | 0.60 | | |
| Ba | 0.02 | | 0.02 | 280 | 0.03 | | 0.02 | 26.3 | 1.00 |
| La | | | 0.01 | 1.38 | | | 0.01 | 1.19 | 0.64 |
| Ce | | | | 2.47 | | 0.003 | bdl | 3.34 | 5.13 |
| Pr | | 0.002 | | 0.22 | 0.003 | 0.01 | 0.01 | 0.29 | 0.11 |
| Nd | | | | 0.75 | | | 0.02 | 1.05 | 1.79 |
| Sm | 0.07 | 0.07 | 0.06 | 0.06 | | 0.04 | | 0.28 | 0.20 |
| Eu | | 0.004 | 0.02 | 0.02 | | | | 0.07 | |
| Gd | | 0.09 | 0.01 | 0.07 | | 0.04 | 0.08 | 0.14 | 0.20 |
| Tb | | 0.002 | 0.002 | 0.01 | 0.01 | 0.003 | 0.01 | 0.01 | |
| Dy | | | | 0.08 | | 0.06 | | 0.09 | |
| Ho | 0.002 | | | 0.01 | | | | 0.02 | |
| Er | | | 0.04 | 0.02 | | | | 0.05 | |
| Tm | | | 0.002 | | | | | 0.01 | 0.03 |
| Yb | | 0.01 | 0.01 | | | 0.01 | | 0.04 | 0.19 |
| Lu | | | 0.002 | | 0.004 | 0.01 | | 0.01 | bdl |
| Hf | 0.05 | 0.06 | 0.04 | 0.30 | 0.03 | 0.03 | 0.03 | 0.11 | 0.15 |
| Ta | 0.02 | 0.03 | 0.03 | 1.32 | 0.02 | 0.02 | 0.01 | 0.18 | 0.06 |
| Pt | | | | | | 0.08 | 0.02 | 0.12 | |
| Au | | | | | 0.05 | 0.04 | | 0.06 | 0.29 |
| Tl | | | | 0.38 | 0.03 | 0.04 | 0.04 | 0.10 | 0.17 |
| Pb | 0.03 | 0.04 | 0.02 | 0.45 | 0.04 | 0.03 | 0.03 | 0.10 | 0.17 |
| Bi | 0.04 | 0.04 | 0.04 | 0.11 | 0.05 | 0.05 | 0.04 | 0.07 | 0.14 |
| Ni/Co | 7.47 | 7.70 | 7.60 | 7.13 | 7.81 | 7.96 | 8.25 | 8.84 | 8.71 |

| | SN-N6 | | | | | | SN-N9 | | |
|---|---|---|---|---|---|---|---|---|---|
| | | | rim | | resorbed | resorbed | | secondary | secondary |
| | Dunite | | | | | | Dunite | | |
| Sc | 1.79 | 2.09 | 1.61 | 3.28 | 4.69 | 4.25 | 1.56 | 1.50 | 2.77 |
| Ti | 7408 | 7318 | 7475 | 8182 | 9301 | 9311 | 7560 | 10795 | 6605 |
| V | 532 | 533 | 568 | 576 | 559 | 601 | 1203 | 1091 | 1144 |
| Mn | 955 | 951 | 1045 | 935 | 1085 | 1180 | 1519 | 2503 | 2570 |
| Co | 217 | 233 | 233 | 225 | 225 | 235 | 278 | 208 | 283 |
| Ni | 1379 | 1390 | 1591 | 1378 | 1442 | 1501 | 1804 | 1344 | 1611 |
| Cu | 7.32 | 4.56 | 7.86 | 4.35 | 2.89 | 7.04 | 1.72 | 4.01 | 3.94 |
| Zn | 563 | 564 | 603 | 535 | 565 | 655 | 901 | 877 | 1085 |
| Ga | 65.8 | 65.3 | 76.8 | 68.5 | 60.7 | 66.6 | 83.9 | 40.1 | 62.5 |
| Se | 2.30 | 2.56 | 2.53 | 2.11 | 2.88 | 2.46 | 6.34 | 11.3 | 10.5 |
| Rb | 0.10 | 0.11 | 0.13 | 1.34 | 0.98 | 0.97 | 0.36 | 28.1 | 0.71 |
| Sr | 0.03 | 0.04 | 0.03 | 6.30 | 8.27 | 20.8 | 0.07 | 27.2 | 0.15 |
| Y | | 0.01 | | 0.20 | 0.29 | 0.53 | 0.08 | 0.07 | 0.10 |
| Nb | 0.40 | 0.36 | 0.42 | 1.67 | 1.69 | 9.56 | 1.72 | 76.6 | 1.65 |

**Table 4.** *Cont.*

| | | | SN-N6 rim Dunite | | resorbed | resorbed | | SN-N9 secondary Dunite | secondary |
|---|---|---|---|---|---|---|---|---|---|
| Zr | 0.76 | 0.63 | 0.78 | 2.17 | 3.17 | 7.19 | 2.43 | 31.9 | 3.29 |
| Ag | 0.07 | 0.10 | 0.05 | 0.08 | 0.05 | 0.03 | 0.06 | 0.16 | 0.29 |
| Cd | 0.24 | 0.25 | 0.26 | 0.22 | 0.09 | 0.35 | 0.75 | 1.36 | 1.61 |
| In | 0.02 | 0.03 | 0.02 | 0.02 | 0.04 | 0.04 | 0.12 | 0.07 | 0.07 |
| Sn | 0.52 | 0.76 | 0.48 | 0.70 | 0.70 | 0.82 | 2.15 | 3.18 | 3.70 |
| Sb | 0.32 | 0.28 | 0.26 | 0.22 | 0.26 | 0.23 | 0.59 | 0.69 | 0.77 |
| Te | 0.71 | 1.01 | 0.95 | 0.64 | 0.65 | 0.74 | | | |
| Ba | | 0.05 | | 20.0 | 12.0 | 77.8 | | 33.9 | 0.33 |
| La | 0.01 | | | 0.75 | 0.61 | 2.61 | 0.06 | 2.40 | |
| Ce | | 0.003 | 0.003 | 1.57 | 1.31 | 4.99 | | 3.39 | 0.07 |
| Pr | | | | 0.14 | 0.15 | 0.52 | | 0.32 | |
| Nd | 0.07 | | 0.08 | 0.55 | 0.55 | 1.39 | | 1.36 | 0.44 |
| Sm | 0.09 | | | 0.06 | 0.16 | 0.26 | | 0.09 | |
| Eu | 0.02 | 0.004 | 0.02 | 0.03 | 0.04 | 0.05 | | 0.04 | 0.09 |
| Gd | 0.13 | 0.02 | 0.02 | 0.06 | 0.10 | 0.15 | | 0.25 | 0.60 |
| Tb | | 0.01 | 0.003 | 0.01 | 0.01 | 0.02 | | 0.01 | 0.02 |
| Dy | | 0.01 | | 0.05 | 0.08 | 0.16 | | 0.05 | 0.07 |
| Ho | | 0.01 | 0.003 | 0.01 | 0.01 | 0.02 | 0.05 | 0.03 | |
| Er | | 0.01 | 0.02 | 0.04 | 0.04 | 0.04 | | | 0.17 |
| Tm | | bdl | 0.003 | 0.002 | 0.002 | 0.003 | 0.02 | 0.06 | |
| Yb | | | | 0.01 | 0.01 | | 0.09 | 0.11 | |
| Lu | 0.003 | 0.01 | | 0.002 | 0.002 | | bdl | | 0.05 |
| Hf | 0.04 | 0.04 | 0.08 | 0.13 | 0.08 | 0.16 | 0.27 | 0.61 | 0.06 |
| Ta | 0.05 | 0.04 | 0.03 | 0.11 | 0.09 | 0.54 | 0.18 | 4.17 | 0.17 |
| Pt | 0.01 | 0.07 | | 0.06 | | 0.03 | | 0.26 | 0.12 |
| Au | | 0.04 | | 0.01 | | 0.01 | | | 0.17 |
| Tl | 0.06 | 0.05 | 0.06 | 0.03 | 0.06 | 0.05 | 0.11 | 0.09 | 0.15 |
| Pb | 0.04 | 0.04 | 0.05 | 0.09 | 0.06 | 0.74 | 0.07 | 0.29 | 0.08 |
| Bi | 0.05 | 0.05 | 0.05 | 0.05 | 0.05 | 0.05 | 0.13 | 0.10 | 0.17 |
| Ni/Co | 6.36 | 5.96 | 6.82 | 6.14 | 6.41 | 6.39 | 6.50 | 6.47 | 5.69 |

Due to the relatively large size of the spinel grains analyzed, LA-ICP-MS data represent clean spinel material in most cases (Figure 2). Only rarely, rim spinels and resorbed spinel grains contain silicate material. Because of LA method limitations, the presence of such material can affect analyzed concentrations of the elements such as Ti, Ba, high field strength elements (HFSE) and the REE (Figure 8). That can compromise a real composition of the analyzed spinel grains. In one case (harzburgite SN-N18), we measured compositions of silicate glass within the completely resorbed spinel grain. The glass corresponds to high-Mg subalkaline basalt with 45–49 wt% $SiO_2$, 2.4–3.6 wt% $TiO_2$, 6.4–9.6 wt% $Al_2O_3$, 0.9–1.4 wt% $Cr_2O_3$, 5.2–5.3 wt% FeO, 12.8–14.7 wt% MgO, 21.4–21.6 wt% CaO and 0.4–0.5 wt% $Na_2O$ and contains 71–99 ppm Sc, 263–281 ppm V, 288–312 ppm Mn, 23–30 ppm Co, 209–241 ppm Ni, 0.7–1.4 ppm Cu, 36.8–72.7 ppm Zn, 18–20 ppm Ga, 1.2–4.1 ppm Rb, 106–155 ppm Sr, 12–19 ppm Y, 5.6–13 ppm Nb, 152–201 ppm Zr, 13–30 ppm Ba, 105–160 ppm total REE, 8.2–8.9 ppm Hf, 1.3–2.7 ppm Ta and 0.3–1.4 ppm Pb.

3.2.1. Spinel in Cpx-Poor Spl Lherzolites

Spinels of the two major types (Figure 3a,b) display slightly different trace element compositions. Primary grains are characterized by concentrations of Sc 1.1–1.4 ppm, Ti 175–217 ppm, V 674–755 ppm, Mn 693–831 ppm, Ni 1912–2071 ppm, Co 218–240 ppm, Cu 3.9–5.7 ppm, Zn 641–691 ppm and Ga 40–44 ppm. The Ni/Co ratio is 8.4–9.2. Concentrations of other trace elements are very low (<< 1 ppm and are mostly bdl), although Nb (0.20–0.39 ppm), Zr (0.15–0.21 ppm) and Sn (0.11–0.40 ppm) display somewhat elevated concentrations. Some insignificant compositional variations could be suggested in the

direction from the central to marginal parts of the primary spinel grains. The marginal parts of the grains are characterized by slightly lower concentrations of Mn, Co, Ni, Zn and Ga.

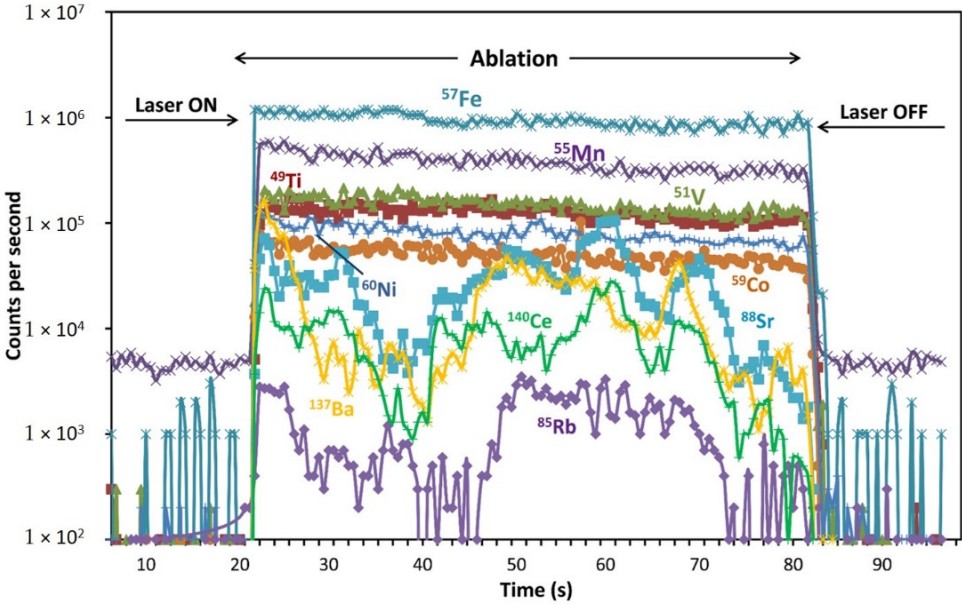

**Figure 8.** An example of time-resolved LA-ICP-MS depth profile (dunite sample SN-N6; resorbed Spl grain; beam diameter is 35 μm) showing uneven Rb (isotope $^{85}$Rb), Sr ($^{88}$Sr), Ba ($^{137}$Ba) and Ce ($^{140}$Ce) distribution across spinel grain volume due to the incorporation of silicate material into the resorbed spinel grain. From the left, the background count is 22 s (not shown completely), followed by 60 s of the ablation time, which is integrated and by 40 s of the wash-out time (not shown completely). A few selected isotopes with even distribution (which are not abundant in silicate glass material) are shown for comparison.

The spinel rim displays lower concentrations of Sc (bdl to 0.36 ppm), Co (191–208 ppm) and Ni (1456–1556 ppm) and higher concentrations of V (893–924 ppm) and Mn (828–889 ppm) than primary spinel. The Ni/Co ratio (7.5–8.7) is lower than that for the primary spinel grains. Concentrations of Cu (4.3–4.7 ppm), Zn (679–693 ppm) and Ga (40–44 ppm) are similar to those in the primary spinels. Concentrations of other trace elements are very low (<< 1 ppm) and are mostly below the detection limits (except for Nb with the concentrations of 0.22–0.36 ppm).

### 3.2.2. Spinel in Cpx-Rich Spl Lherzolites

Spinels of the two types identified in Cpx-rich spinel lherzolites (Figure 4a,c) are characterized by somewhat different trace element compositions. Primary spinel represented by clean grains is homogeneous and is characterized by the concentrations of Sc of 0.73–0.91 ppm, Ti of 1264–1356 ppm, V of 489–534 ppm, Mn of 618–676 ppm, Ni of 2539–2815 ppm, Co of 204–222 ppm, Cu of 2.8–3.3 ppm, Zn of 538–620 ppm and Ga of 66–74 ppm. The Ni/Co ratio is high (12.4–12.7). Concentrations of other trace elements are very low (<< 1 ppm and are often bdl), but Nb (0.11–0.16. ppm), Zr (0.09–0.19 ppm) and Sn (0.19–0.30 ppm) display somewhat elevated concentrations.

The spinel rim is characterized by higher concentrations of Sc (2.1–2.9 ppm) and V (630–699 ppm), lower concentrations of Ti (963–1121 ppm), Co (169–214 ppm) and Ni (1636–2131 ppm) and similar Mn (610–649 ppm), Cu (3.2–3.3 ppm), Zn (499–600 ppm) and Ga (60–70 ppm). The Ni/Co ratio is much lower than in the primary spinel (9.6–10.0) and is similar to that in spinels from Cpx-poor spinel lherzolites. Concentrations of other trace elements are very low (<< 1 ppm) and are often below the detection limits (including Nb, Zr and Sn).

Trace element composition of spinels from the sample XLT-5 (Figure 4b) is not so different as compared to other spinels from the lherzolite matrix. These low-Cr spinels are characterized by lower concentrations of Sc (from bdl to 0.30 ppm), Ti (172–216 ppm), V (348–397 ppm) and Cu (0.23–0.81 ppm) and higher concentrations of Zn (1410–1637 ppm) and Co (435–470 ppm) as compared to other primary lherzolite spinels. Concentrations of Mn (639–694 ppm), Ni (2920–3188 ppm) and Ga (69–79 ppm) are similar to those in other spinels of the lherzolite matrix. The ratio of Ni/Co (6.5–6.8) is slightly lower than that in other primary lherzolite spinels. Concentrations of other trace elements are low (mostly < 1 ppm). No differences between the trace element composition of the core and the edge of the spinel grain were observed.

### 3.2.3. Spinel in Spl-Grt Lherzolites

Spinels of three types identified in the Spl-Grt lherzolites (Figure 5a–c) are characterized by varying trace element compositions. Primary spinel represented by clean grains is compositionally homogeneous and contains 0.73–1.7 ppm of Sc, 1044–2900 ppm of Ti, 536–742 ppm of V, 625–809 ppm of Mn, 212–229 ppm of Co, 2347–2936 ppm of Ni, 593–748 ppm of Zn, 1.8–5.7 ppm of Cu and 69–112 ppm of Ga. Such spinels are characterized by the high Ni/Co ratios (10.9–12.9). Concentrations of other trace elements are very low (<<1 ppm) and are often below the detection limits. However, Nb (0.09–0.21 ppm), Zr (0.12–0.30 ppm) and Sn (0.20–0.40 ppm) display elevated concentrations.

The rim surrounding primary spinel grains can often replace smaller grains completely, producing resorbed grains. Because of the textural features, trace element composition of the spinels of this type was sometimes compromised by the incorporation of silicate material (see Figure 8). This spinel is characterized by higher concentrations of Ti (1549–1654 ppm), lower concentrations of Co (186–188 ppm) and Ni (1557–2188 ppm) and similar to the primary spinel concentrations of V (616–1216 ppm), Mn (880–1052 ppm), Zn (646–746 ppm) and Ga (74–106 ppm). The secondary spinels display slightly lower Ni/Co ratios (8.3–11.7) than the primary spinels. Concentrations of other trace elements (including Sc, Cu, Nb, Zr and Sn) are very low (<<1 ppm) and are often below the detection limits.

Spinels from the kelyphite rim replacing garnet are characterized by slightly higher concentrations of Sc (2.60–4.27 ppm) and Mn (1412–1841 ppm) as compared to primary spinels of the lherzolite matrix. On the other hand, such spinels display distinctly lower concentrations of Ti (156–217 ppm), V (196–251 ppm), Zn (51–64 ppm), Ga (10–14 ppm), Ni (223–260 ppm) and Co (116–132 ppm). The Ni/Co ratio is very low (1.92–1.97). In higher concentrations of Sc and Mn and lower concentrations of Ti, V, Ni, Co, Zn and Ga, kelyphite spinels are different from otherwise similar (especially in terms of the major oxides) low-Cr spinels from the Cpx-rich spinel lherzolite XLT-5. Concentrations of most other trace elements in kelyphite spinel are very low (<<1 ppm) and are often below the detection limits.

### 3.2.4. Spinel in Harzburgites

Spinels of three types identified in harzburgites (Figure 6) are characterized by somewhat varying trace element compositions. Primary spinel is homogeneous and is characterized by concentrations of Sc of 0.76–1.4 ppm, Ti of 5061–6544 ppm, V of 588–669 ppm, Mn of 752–871 ppm, Zn of 687–808 ppm, Ga of 124–148 ppm, Ni of 1572–1688 ppm and Co of 197–222 ppm. The Ni/Co ratio (7.5–8.4) is much lower than that in primary lherozilte spinels. A rim surrounding some harzburgite spinel grains contains more Sc (2.99 ppm), Ti (9251 ppm), Mn (1066 ppm), Ni (1956 ppm), Co (233 ppm) and Zn (847 ppm) than the primary spinel and similar amounts of V (652 ppm) and Ga (151 ppm). The rim displays Ni/Co ratio of 8.8.

Irregular resorbed interstitial spinel grains are characterized by the composition which is significantly different from that of either primary spinel or the rim around the primary grains. It displays higher concentrations of Sc (3.1–3.5 ppm), Ti (17,159–18,024 ppm), V (847–998 ppm), Mn (1563–1669 ppm), Ni (169–1933 ppm), Co (222–237 ppm) and Bi

(0.11–0.15 ppm) than both primary spinel and the rim. On the contrary, concentrations of Ga (86–117 ppm) are lower than in both primary spinel and the rim. Concentrations of Zn vary from 687 to 878 ppm covering the whole range of Zn concentrations observed in all types of harzburgite spinel. Concentrations of Co (179–237 ppm) and Cu (2.03–2.65 ppm) are similar in spinels of all three identified types. The Ni/Co ratio (7.1–7.4) is slightly lower in the resorbed spinel than in the primary spinel. The rest of the trace elements display similarly low concentrations (<1 ppm) in spinel of all three identified textural types. Because of the textural features of the rim and resorbed spinels, analyzed trace element composition might have been compromised by the incorporation of silicate material (see Figure 8).

### 3.2.5. Spinel in Dunites

Spinel of several types identified here (Figure 7a–e) varies in concentrations of the trace elements both within a single sample and between the two studied dunite samples (SN-N6 and SN-N9). In sample SN-N6 (Figure 7a–c), primary spinel represented by clean homogeneous grains is characterized by the concentrations of Sc of 1.6–2.2 ppm, Ti of 7318–7511 ppm and Cu of 4.6–7.9 ppm. A rim developed around some primary spinel grains displays higher concentrations of Sc (3.3 ppm) and Ti (8182 ppm) and slightly lower concentrations of Cu (4.4 ppm) than the primary grain. Concentrations of Mn (935–1045 ppm) are similar in both primary grains and the rim. Irregular resorbed interstitial spinel grains are characterized by high concentrations of Sc (4.1–4.7 ppm), Ti (9213–9397 ppm) and Mn (1073–1180 ppm). Concentrations of V (532–604 ppm), Zn (535–655 ppm), Ni (1378–1591 ppm), Co (217–233 ppm), Ga (60–77 ppm) and Se (2.1–3.8 ppm) are similar in spinels of all types. The Ni/Co ratio is low (5.8–6.5) and is similar for both primary spinel and spinel from the rim. Because of the textural features of the rim spinels, analyzed trace element composition could be compromised by the incorporation of silicate material to the resorbed grains/rim (see Figure 8).

Dunite sample SN-N9 contains interstitial spinel grains of two types (Figure 7d,e). Spinel of the first type (texturally similar to the primary spinel in dunite sample SN-N6) contains significantly higher amounts of V (1203 ppm), Mn (1519 ppm), Ni (1904 ppm), Co (278 ppm), Zn (901 ppm), Ga (84 ppm), Se (6.3 ppm), Zr (2.43 ppm) and Bi (0.13 ppm) than the primary spinel in sample SN-N6. On the other hand, concentration of Cu is lower (1.7 ppm) than in sample SN-N6. Spinel of the second type in the sample SN-N9 (absolutely atypical for other samples irregular to skeletal grains in altered silicate matrix) is characterized by varying concentrations of Sc (1.50–2.77 ppm), Ti (6605–10,795 ppm), Cu (3.9–20 ppm) and Zn (877–1085 ppm), high concentrations of Se (10–11 ppm) and moderately low concentrations of Ga (40–63 ppm). Concentrations of Co (208–278 ppm) and Ni (1344–1804 ppm) are overall similar in spinels of all types from the two dunite samples and the Ni/Co ratio is low (5.7–6.5) i.e., similar to that for spinels from the sample SN-N6.

## 4. Discussion

The East Antarctic SCLM rocks have experienced multiple episodes of infiltration and modification by silicate and minor carbonate and sulfide melts. That resulted in partial melting, recrystallization and generation of metasomatic minerals [4,5,7,19,30,36]. Although spinel is one of the most stable mantle minerals resistant to secondary alterations, it can be modified by cryptic metasomatism through fluid (and/or melt) to rock interaction. This allows spinel to provide information on processes of both depletion and enrichment in the upper mantle.

Differences in chemical compositions of spinel from the upper mantle xenoliths collected in the two intrusions in Jetty Peninsula are pronounced on both the major oxide and trace element levels. Spinels from the xenoliths studied show compositional features which allow us to see differences between the minerals from lherzolite and non-lherzolite peridotites (diagrams $Al_2O_3$ vs. $TiO_2$, Mg# vs. Cr# and $Al_2O_3$ vs. $Cr_2O_3$; Figures 9–11).

For all analyzed spinels, V, Co, Cu, Zn and Mn concentrations increase whereas Ga and Ni concentrations and Ni/Co ratios decrease with increasing Cr/Al ratios (Figure 12). That suggests partial melting of the host peridotite. Overall, we have defined studied upper mantle xenoliths as the residues of 6–12% (Yuzhnoe body) and 8–16% (Severnoe body) of mantle melting (see below).

Higher amounts of $Fe^{3+}$ (and the magnetite component) in both some rims around primary spinels and resorbed spinel grains (Table 1, the Electronic Supplementary Table S1) are likely due to the increasing degrees of spinel alteration due to the interaction with different interstitial melts (see [4,30,37]).

### 4.1. Spinels from Lherzolite Xenoliths

Spinels from Cpx-poor spinel lherzolites are the simplest in terms of their history. Using the equation from [38] (based on the primary spinel Cr#), we have defined Cpx-poor Spl lherzolites from the Yuzhnoe intrusion as residues of 11.0 and 11.8% partial melting. Spinels from Cpx-poor Spl lherzolites are the only spinel type with compositions that fall completely within the field of the mantle peridotites (not influenced by the metasomatic agents) (Figure 9). Compositions of Ti-richer and Al-poorer rims surrounding low-Ti spinel cores (i.e., primary spinels) fall in the area where the mid-ocean ridge basalt (MORB) and mantle peridotite fields overlap. That might suggest metasomatic influence of the MOR-like melts migrating through the peridotite matrix.

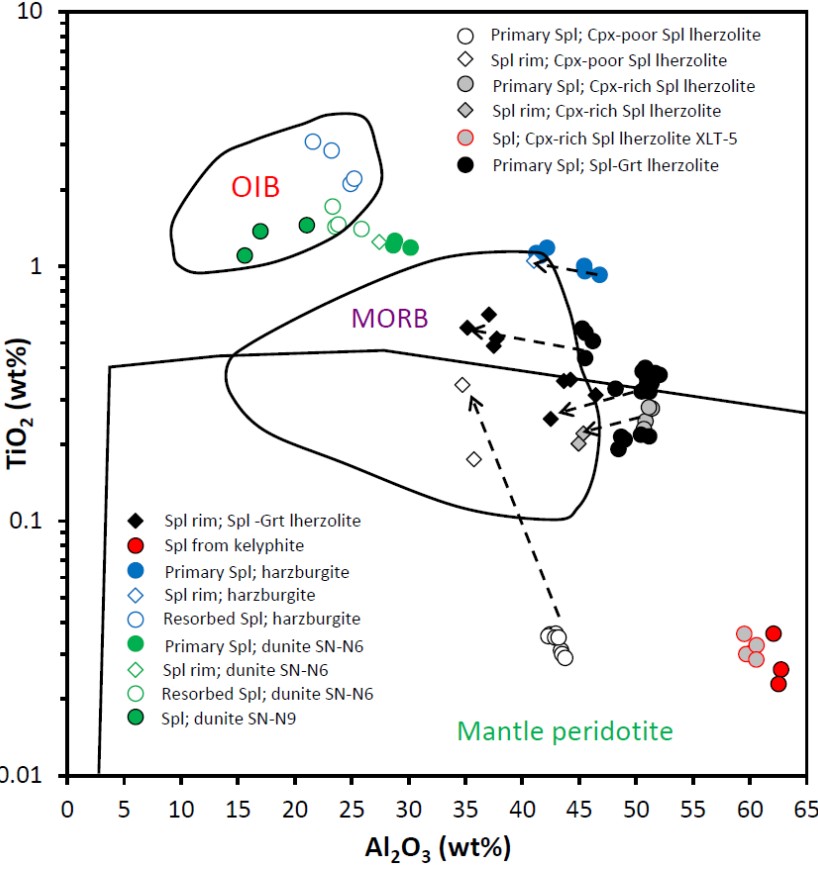

**Figure 9.** Diagram $Al_2O_3$ (wt%) vs. $TiO_2$ (wt%) for spinels from the East Antarctic peridotite xenoliths. Dashed arrows connect compositions of the spinel cores and corresponding rims (see text for details). Fields for Mantle peridotite, mid-ocean ridge basalt (MORB) and ocean island basalt (OIB) spinels are after [18,21,23].

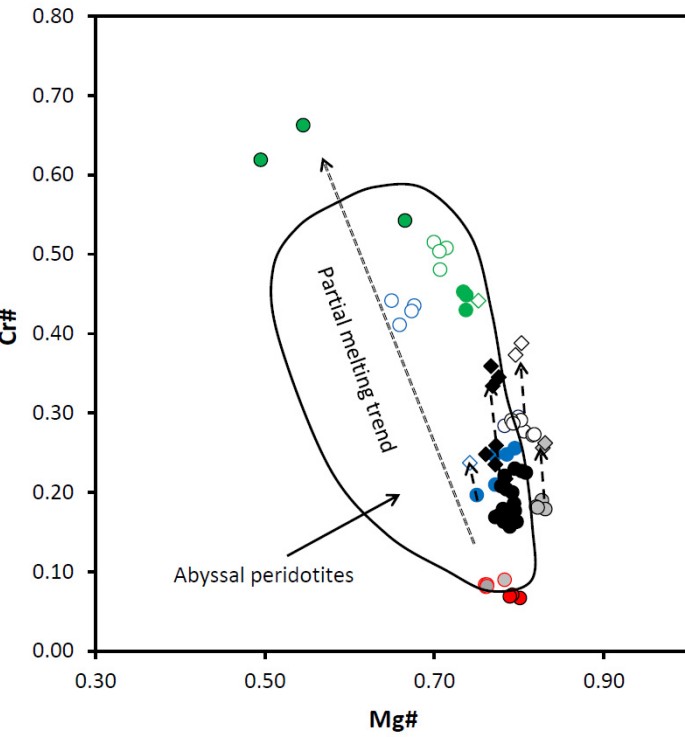

**Figure 10.** Diagram Mg# vs. Cr# for spinels from the East Antarctic peridotite xenoliths. Cr# = [Cr/(Cr + Al)] (at%); Mg# = [Mg/(Mg + Fe$^{2+}$)] (at%); Field for Abyssal peridotites and the Partial melting trend are after [18,24]. Symbols are as in Figure 9.

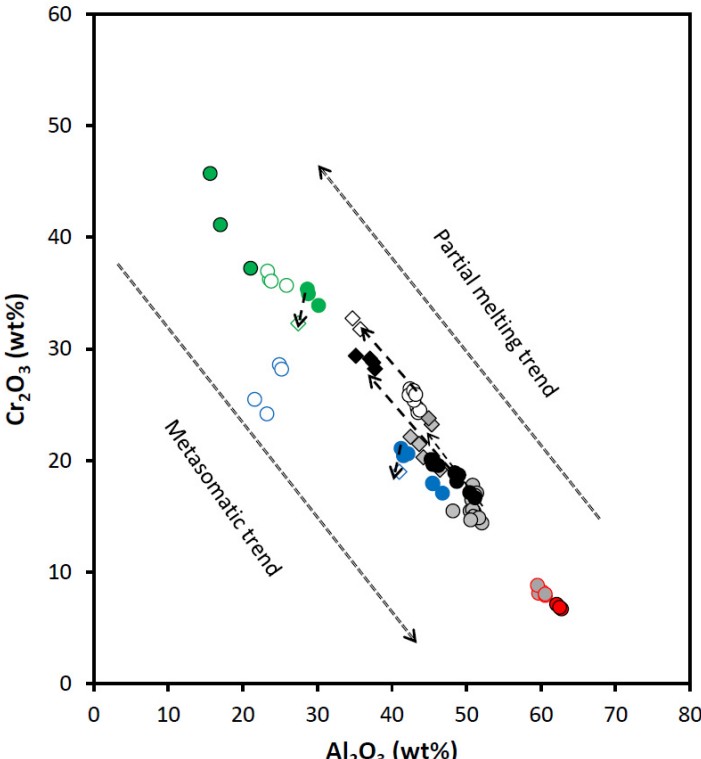

**Figure 11.** Diagram Al$_2$O$_3$ (wt%) vs. Cr$_2$O$_3$ (wt%) for spinels from the East Antarctic peridotite xenoliths. The "primary spinel–spinel rim" composition pairs for harzburgite and dunite deviate strongly from the Partial melting trend (see text for details). Partial melting trend and Metasomatic trend are after [24]. Symbols are as in Figure 9.

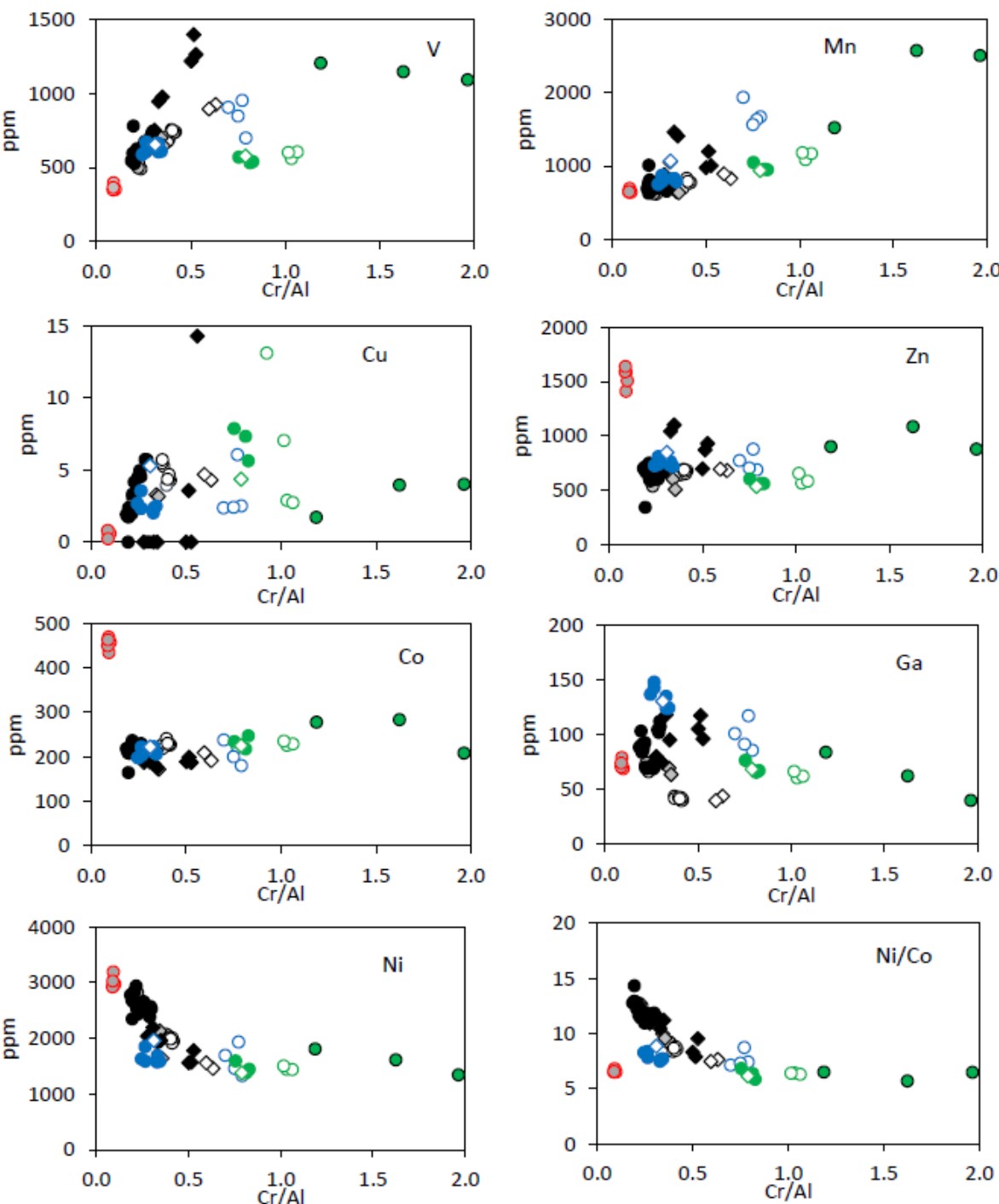

**Figure 12.** Plots of mantle spinel Cr/Al vs. various trace elements and Ni/Co from East Antarctic peridotite xenoliths. Symbols are as in Figure 9.

Spinels from both Cpx-rich Spl and Spl-Grt lherzolites display compositional features which are different from those observed for the Cpx-poor Spl lherzolites. We have defined degrees of melting [38] for Cpx-rich Spl lherzolites as varying between 6.8 and 7.4%. Spl-Grt lherzolites were affected by partial melting at degrees varying between 5.8 and 9.3%. Compositionally, primary spinels from Cpx-rich Spl and Spl-Grt lherzolites fall in the more fertile part of the partial melting trend than spinels from Cpx-poor Spl lherzolites (diagrams Mg# vs. Cr# and $Al_2O_3$ vs. $Cr_2O_3$; Figures 10 and 11). Simultaneous increase in $Cr_2O_3$ concentrations and decrease in $Al_2O_3$ concentrations (Figure 11) suggest that partial

melting may be involved in the generation of both spinel rims and completely resorbed spinel grains. Compositional points of primary spinels from these two types of lherzolites are located in the Ti- and Al-rich part of the mantle peridotite field and in the most Al-rich part of the MORB field in the $Al_2O_3$ vs. $TiO_2$ diagram (Figure 9). Compositions of both the rims and completely resorbed spinel grains fall in the field of the MORB spinels (Figure 9). That suggests influence of the migrating MORB-like melts on the peridotite matrix during generation of spinel rims and completely resorbed spinel grains.

Two major clusters of compositions can be observed in the diagram $TiO_2$ vs. Cr# (Figure 13): one for spinels from Cpx-poor Spl lherzolites and another one for spinels from Cpx-rich Spl and Spl-Grt lherzolites. Compositions of primary spinels from Cpx-poor Spl lherzolites do not fall far away from the "Fertile MORB mantle partial melting" trend, suggesting only an insignificant influence of the processes other than partial melting. Compositions of spinel rims from Cpx-poor Spl lherzolites differ significantly from those of primary spinels. Insignificant increase in Cr# accompanied by significant increase in $TiO_2$ concentrations (Figure 13) suggests that melting did not play a major role in generation of such rims. On the contrary, interaction with the MORB-like melt could be suggested as a preferable way to form the rims (although some melting might have taken place during the spinel-melt interaction; see [4]).

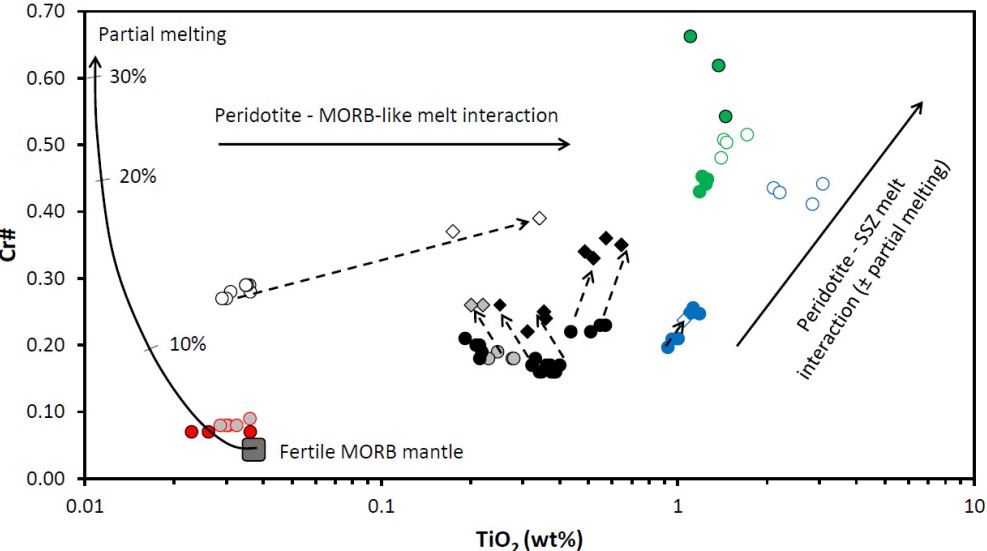

**Figure 13.** Diagram $TiO_2$ (wt%) vs. Cr# for spinels from the East Antarctic peridotite xenoliths. "Fertile MORB mantle partial melting" trend, "Peridotite–MORB-like melt interaction" trend and "Peridotite–SSZ melt interaction" trend are after [28]. Ticks on the "Fertile MORB mantle partial melting" trend are degrees of the partial melting and melt extraction (after [28,39]). Symbols are as in Figure 9.

Compositions of spinels from Cpx-rich Spl and Spl-Grt lherzolites fall aside from the "Fertile MORB mantle partial melting" trend generally forming a horizontal cluster in Figure 13. That suggests that processes other than melting were mostly responsible for generation of what we now classify as primary spinels. Most likely, interaction of the host lherzolite matrix with the MORB-like melt was responsible for change in the composition of spinel cores (increase in $TiO_2$ concentrations at constant Cr#) originally formed during the insignificant (6–9%) partial melting of the host lherzolite. Both generation of the rim spinels and complete resorption of some spinel grains can be due to the two major processes: (i) partial melting and recrystallization of primary spinel (for both Cpx-rich Spl and Spl-Grt lherzolites) as it can be judged from increase in Cr # at decrease in $TiO_2$ concentrations and (ii) later interaction of such spinels with the suprasubduction zone (SSZ)-like melts (spinels from Spl-Grt lherzolite DN-4).

In terms of the trace elements, primary spinels from Cpx-poor Spl lherzolites display more complex features than those identified on the basis of the major oxide compositions alone. It is seen in the diagram $Fe^{3+}$# vs. Ga (Figure 14) that $Fe^{3+}$# [$Fe^{3+}/(Fe^{3+} + Al + Cr)$] decreases in the direction from spinel cores to rims at a constant concentration of Ga. Such a feature might be due to the interaction between the mantle lithospheric peridotite and the MORB-like melt (see [28]). Since some primary spinel compositions lie on the arrow connecting spinel core and rims, it is possible that the prolonged interaction between the SCLM rocks and the MORB-like melt could lead to complete recrystallization of genuine primary spinels and a change of their composition.

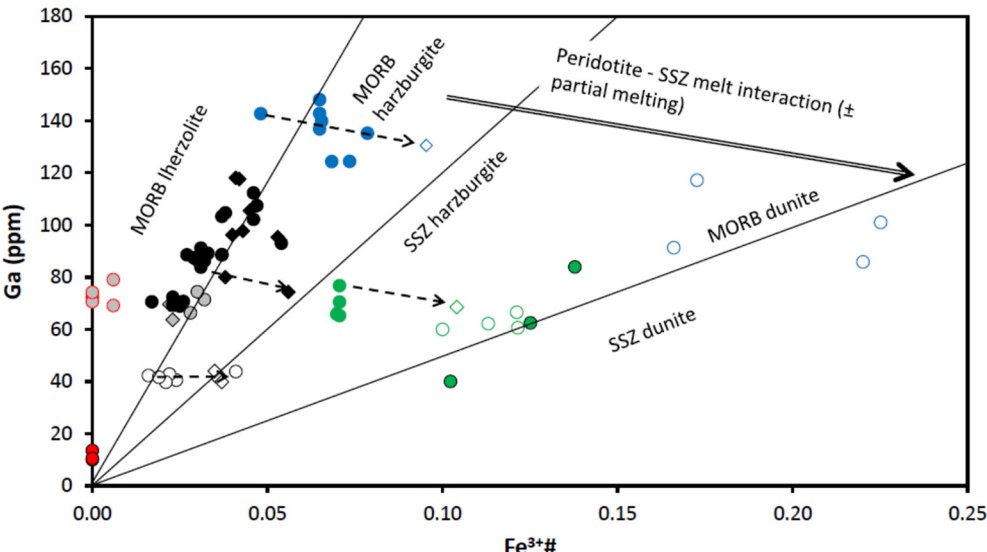

**Figure 14.** Diagram $Fe^{3+}$# vs. Ga (ppm) for spinels from the East Antarctic peridotite xenoliths. Fields for mid-ocean ridge basalt (MORB) lherzolite, MORB harzburgite, MORB dunite, suprasubduction zone (SSZ) harzburgite and SSZ dunite and "Peridotite–SSZ melt interaction" trend are after [27,28]. Symbols are as in Figure 9.

A different origin of (primary) spinels from samples U-3/4-2 and U-1/4-3 could be suggested from the features seen in Figure 15. While compositions of primary spinels from samples U-3/4-2 are located in the field of the MORB-residual compositions, compositions of rimless spinel cores from the sample U-1/4-3 are located in the field of the SSZ residual compositions. Spinel rims in sample U-3/4-2 may have been mainly generated by the interaction of mantle rocks with the MOR-like melts because their compositions are located in the field of the MORB reactive compositions (Figure 15).

Diagrams involving Ga-Ti-$Fe^{3+}$ systematic (see [28]) allow to discriminate between spinels from Cpx-rich Spl and Spl-Grt lherzolites. Compositions of primary spinels from Cpx-rich Spl lherzolite sample UN-1 fall in the field of the MOR lherzolites in the diagram $Fe^{3+}$# vs. Ga (Figure 14) and in the MORB reactive lherzolites in the diagram Ga/$Fe^{3+}$# vs. 100$TiO_2$/$Fe^{3+}$# (Figure 15). In this, primary spinels from Cpx-rich Spl lherzolite are similar to spinels from Spl-Grt lherzolites. However, spinel rims in lherzolites of these two types display different compositional features. Spinel rims from Cpx-rich Spl lherzolites show a simultaneous decrease in concentrations of Ga and in values of $Fe^{3+}$# (Figure 14). That can be explained by the partial melting, possibly because of the influence of percolating silicate melts (see [4,28]). Spinel rims from Spl-Grt lherzolites, on the other hand, display a more complex relation between Ga concentrations and $Fe^{3+}$# values (see below).

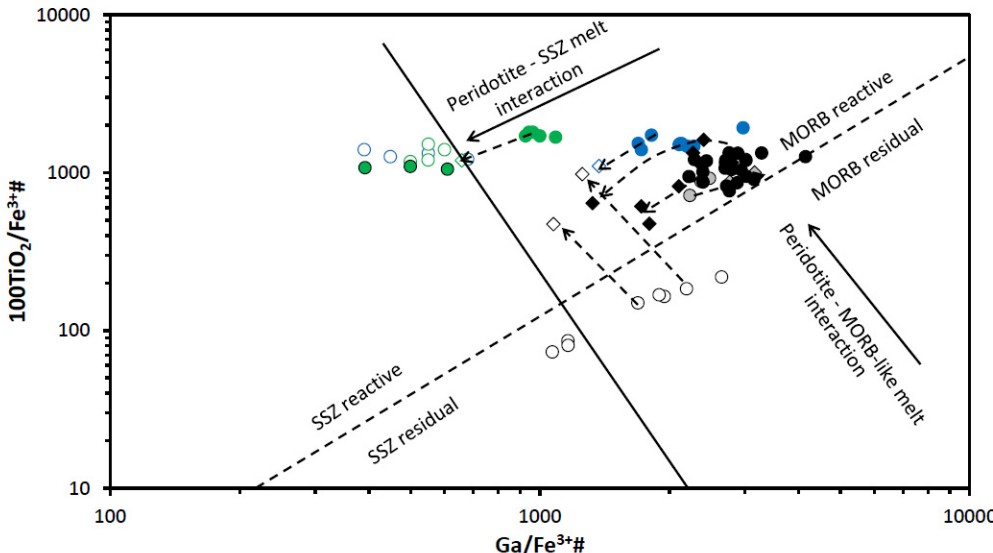

**Figure 15.** Diagram $Ga/Fe^{3+}\#$ vs. $100TiO_2/Fe^{3+}\#$ for spinels from the East Antarctic peridotite xenoliths. Fields for mid-ocean ridge basalt (MORB) residual, MORB reactive, suprasubduction zone (SSZ) residual and SSZ reactive peridotites and "Peridotite–MORB-like melt interaction" trend and "Peridotite–SSZ melt interaction" trend are after [27,28]. Symbols are as in Figure 9.

Spinels from Spl-Grt lherzolites, as suggested above, were likely resulted firstly from partial melting of the host lherzolite and then from interaction with various melts. If lower $Ga/Fe^{3+}\#$ ratios could point to the interaction of peridotite matrix with the SSZ-related melts, increase in such ratios could suggest interaction with the MORB-like melts (see [27,28]). It can be suggested from the features seen in the diagram $TiO_2$ vs. Cr# (Figure 13) that some spinel rims resorbed spinels might have been generated because of the influence of partial melting, whereas others resulted from the interaction with the SSZ-like melts (not excluding influence of partial melting though). Similar conclusions could be made on the basis of the features observed in the diagram $Fe^{3+}\#$ vs. Ga (Figure 14). The diagram $Ga/Fe^{3+}\#$ vs. $100TiO_2/Fe^{3+}\#$ (Figure 15) suggests a more complex origin of the spinel rims and resorbed spinels, which could include partial melting, interaction with the MORB-like melts and finally, interaction with the SSZ-related melts.

In all diagrams considered above, compositions of spinels from the kelyphite rims (Spl-Grt lherzolites) and spinels from lherzolite matrix of the Cpx-rich Spl lherzolite xenolith XLT-5 fall significantly away from all other spinel compositions. That points to a different origin of spinels of these two types. Kelyphite rims could be formed as a result of isochemical garnet decomposition under the changing P-T conditions (development of the rift and migration of the spinel-garnet transition towards the deeper mantle horizons in the case of the studied peridotites) [4,40]. Spinels from the kelyphite, therefore, are a by-product of the garnet break-down.

Low-Cr and high-Al spinels from the Cpx-rich Spl lherzolite xenolith XLT-5 were first described by [4]. Those authors calculated a low ambient temperature (834 °C) for the rock. They suggested that the lherzolite of this type initially belonged with the cold harzburgite protolith whose refertilization resulted in the addition of clinopyroxene. Therefore, spinel XLT-5 (although similar to kelyphite spinel in many compositional features) probably represents a relic mineral from the shallow cold protolith not affected by refertilization processes yet. This is consistent with low concentrations of the elements such as Sc, Ti, V and Cu (Table 3 and the Electronic Supplementary Table S2).

*4.2. Spinels from Non-Lherzolite Xenoliths*

Spinels from harzburgite display a much wider range of trace element compositions than any analyzed lherzolite spinel (Tables 3 and 4 and the Electronic Supplementary Table S2).

Applying equation from [38] to the primary harzburgite spinel compositions, we have defined harzburgite xenoliths as residues of 7.7–10.4% partial melting. Compositions of primary harzburgite spinels are located very close to the field of the MORB spinels in Figure 9. Points of harzburgite spinel compositions fall completely into the field of the abyssal peridotites in Figure 10 and also lay along the partial melting trend in Figure 11. On the other hand, compositions of spinel rims and completely resorbed spinel grains suggest the influence of the SSZ-like melts to various extents (Figures 9 and 13). A diagram $Cr_2O_3$ vs. $Al_2O_3$ (Figure 11) provides even more evidence of the SSZ-related melt influence on the host peridotite. Interaction between the mantle peridotites and slab-derived fluids (melts) may produce a strong heterogeneity that modifies both the Al and Cr contents (i.e., Cr#) of primary spinel [19,24] and therefore produces a reverse (metasomatic) trend in Cr# values [24]. A composition of harzburgite spinel rim does not fall on the partial melting trend, but rather tends to move toward the reverse trend (Figure 11).

As it is seen in the diagram $Fe^{3+}$# vs. Ga (Figure 14), compositions of primary harzburgite spinels fall in the field of the MORB harzburgite, i.e., the host rocks might have been affected by interaction with the MORB-like melts (as was already suggested on the basis of the major oxide compositions). Similar suggestions can be made from the features observed in the diagram $Ga/Fe^{3+}$# vs. $100TiO_2/Fe^{3+}$# (Figure 15). Unlike primary spinels, spinel rims and completely resorbed spinel grains might have been significantly modified by interaction with the SSZ-related melts (diagrams $Fe^{3+}$# vs. Ga; Figure 14 and $Ga/Fe^{3+}$# vs. $100TiO_2/Fe^{3+}$#; Figure 15). Such interaction could be so significant that some resorbed harzburgite spinels display compositions corresponding to spinels from SSZ dunites (Figure 14).

Spinels in dunite display a significant compositional difference between the two analyzed samples (SN-N6 and SN-N9). Applying the equation from [38] to the composition of primary spinels from dunite xenoliths, we have defined dunites from the Severnoe intrusion as residues of 15.6–17.9% partial melting. Compositionally, spinels from dunite samples stay significantly apart from spinels in any other studied peridotite sample (Tables 3 and 4 and Electronic Supplementary Tables S1 and S2). All dunite spinel compositions are located either between the MORB and ocean island basal (OIB) fields or completely within the OIB field (OIB is a compositional analogue of the SSZ-like melts [41]) in Figure 9. A diagram Mg# vs. Cr# (Figure 10) shows that dunite spinel compositions are located at the most depleted end of the partial melting trend. The most depleted compositions are displayed by spinel from the sample SN-N9. A diagram Cr# vs. $TiO_2$ (Figure 13) also suggests that primary dunite spinel (SN-N6) was likely generated in peridotite that experienced high degrees of partial melting and was then affected by interaction with the MORB-like melts. Further on, these spinels were significantly affected by the interaction with the SSZ-like melts that resulted in generation of the rims and resorbed spinel grains. It is seen in diagram $Al_2O_3$ vs. $Cr_2O_3$ (Figure 11) that the composition of the rim, similarly to the spinel rim from harzburgite, does not fall on the partial melting trend, but rather tends to move toward the reverse (metasomatic) trend. As it is seen in the diagram $Fe^{3+}$# vs. Ga (Figure 14), compositions of primary spinels from dunite SN-N6 fall in the area transitional between the fields of SSZ harzburgite, the MORB dunite and then the SSZ dunite. This is consistent with strong influence of different melts on the host peridotite and, in particular, the SSZ-related melts. As seen in the diagram Ga vs. $Fe^{3+}$# (Figure 14), compositions of primary spinels from dunite SN-N6 fall in the transitional area between the fields of SSZ harzburgite, the MORB dunite and then the SSZ dunite. This is consistent with strong influence of different melts on the host peridotite and, in particular, the SSZ-related melts. A diagram $Ga/Fe^{3+}$# vs. $100TiO_2/Fe^{3+}$# (Figure 15) clearly shows that while primary spinel composition from dunite SN-N6 falls in the MORB reactive field, the rest of the dunite spinel compositions are located in the SSZ reactive field. This may indicate that dunite spinels resulted from partial melting and further interaction with the MORB-like melts were later significantly modified by the SSZ-related melts.

Although primary spinel from dunite sample SN-N9 is compositionally very similar to the resorbed spinel grains from the sample SN-N6, structurally it is much closer to the primary spinel from the sample SN-N6. On the other hand, secondary (with up to skeletal textures; Figure 7e) spinel from the sample SN-N9 is characterized by the highest Cr# at the lowest Al and Mg amounts and contains the highest amounts of Ag and Cd among all studied spinels (Tables 2 and 4 and the Electronic Supplementary Tables S1 and S2). In the diagram $Al_2O_3$ vs. $TiO_2$ (Figure 9) compositional points of these spinels are completely located in the OIB field. These spinels display the most "extreme" compositions in various diagrams. It can be suggested therefore that it represents either a product of the complete recrystallization of spinels of the earlier generation(s) during interaction with the SSZ-like melts infiltrating through the peridotite matrix, or was crystallized directly from such melts.

*4.3. Petrologic Implications from Spinel Compositions*

Because a present-day tectonic position of the studied Antarctic region is determined by the processes of arc accretion and collision (or continental collision) and actively developing rifting [4,6], mantle rocks beneath the Jetty Peninsula area could have interacted with the melts produced both in the rift environments (MORB-like melts) and in the reactivated buried subduction slab (SSZ-related melts). The composition of the SCLM rocks from beneath the Jetty Peninsula suggests that they experienced reaction and refertilization by incorporating and infiltrating silicate and minor carbonate and sulfide melts [4,5,7]. In order to discriminate between peridotites affected by the MORB-like and by the SSZ-related melts, we used diagrams $TiO_2$ vs. Cr#, $Fe^{3+}$# vs. Ga, $Fe^{3+}$# vs. $TiO_2$ and $Ga/Fe^{3+}$# vs. $100TiO_2/Fe^{3+}$# for spinels (see [27,28]).

On the plot $TiO_2$ vs. Cr# (Figure 13), compositions of peridotite spinels do not fall on the trend suggesting that continuous extraction of basaltic melt from a peridotite source causes a systematic depletion in Ti content of spinel (see [28,41] and references therein). Deviation of spinel compositions from this "Fertile MORB mantle melting" trend to higher Ti is likely due to the melt-residue interaction through reaction or impregnation (assuming that the spinels have equilibrated with the interacting melts; e.g., [28,39,42,43]). For lherzolite xenoliths from the Yuzhnoe intrusion, a systematic increase in Ti content from Cpx-poor Spl lherzolites toward Spl-Grt lherzolites at a not very significant variation in degree of melting (Figure 13) suggests that the MORB-like melts affected deeper SCLM levels (Spl-Grt and Cpx-rich Spl lherzolites) to much wider extend than the shallower SCLM levels (Cpx-poor Spl lherzolites). Composition of the spinel rims from Cpx-poor Spl lherzolites differs significantly from that of the primary spinels. Partial melting likely did not play a significant role in generation of the spinel rims, whereas interaction with the MORB-like melts was a preferable way to generate the rims (although some melting likely took place during spinel-melt interaction [4]). Origin of the rims and resorbed spinel grains in Cpx-rich Spl and Spl-Grt lherzolites is supposed to be more complex and can be due to two main processes: (i) partial melting of the primary spinel (for both Cpx-rich Spl and Spl-Grt lherzolites) and (ii) interaction with the SSZ-related melts. That may suggest that although the SSZ-related melts impregnated deeper SCLM horizons (below 65–70 km), they did not spread wide, probably forming a network of percolating melts (cf. [44,45]). This way, some Spl-Grt lherzolites were affected by the melt-rock interaction whereas others were virtually free of the melt influence.

For spinel from peridotites of the northern upper mantle domain (Severnoe intrusion), the increase in Ti content associates with simultaneous increase in the Cr# (Figure 13), indicating that the interacting metasomatizing agent (melt and/or fluid) may have played a significant role in facilitating higher degrees of partial melting. The metasomatizing agent(s) affected peridotites from the northern mantle domain at much higher extent than it was observed for peridotites from the southern domain. One of the indications for the chemical nature of the metasomatic agent is compositional difference between the spinel cores and rims (see [4]). Completely resorbed spinel grains from the non-lherzolite peridotites display evidence of much more intensive influence of the SSZ-related melts (a very significant

increase of both Cr# values and Ti concentrations) and, in the case of dunite SN-N9, by partial melting likely induced by interaction with the migrating melt and/or fluid (increase in Cr# at the decrease of Ti content).

The Ga-Ti-$Fe^{3+}$ systematics suggest that the studied peridotites represent both simple melt residues and residues strongly influenced by the MORB-like melts. Some peridotites were influenced additionally by the SSZ-related melts. Signatures of such influences can be seen in diagrams $Fe^{3+}$# vs. Ga and Ga/$Fe^{3+}$# vs. 100$TiO_2$/$Fe^{3+}$# (Figures 14 and 15, respectively). Whereas compositional points of peridotite spinels from the Yuzhnoe intrusion (southern mantle domain) mostly fall in the field of the MORB lherzolites, spinel compositions from the Severnoe intrusion (northern domain) stretch from the MORB harzburgite to the MORB dunite and even to the SSZ dunite field (from primary to completely resorbed spinels) in Figure 14. These features suggest that non-lherzolite peridotites experienced very intensive melt to rock interaction, which may have involved both the MORB-like and the SSZ-related melts (reactivated from the buried SSZ slab).

The studied peridotite spinels form clusters of compositional points in the fields of the MORB-residue, MORB-reactivate and SSZ-reactive compositions in the plot Ga/$Fe^{3+}$# vs. 100$TiO_2$/$Fe^{3+}$# (Figure 15). Compositions of lherzolite xenoliths fall in the fields of both the MORB residues and the MORB-reactive peridotites. It is consistent with the suggestion about the complex history of the SCLM rocks beneath the region. Processes of the rock to melt interaction might have induced additional partial melting (see [4]). None of the non-lherzolite peridotite compositions falls in the MORB residual field suggesting that signatures of earlier partial melting events were completely masked by the later metasomatic processes. All non-lherzolite spinel compositions form a trend stretching from the field of the MORB reactive to the field of the SSZ reactive compositions. That suggests that peridotites from the northern (non-lherzolitic) SCLM domain experienced a somewhat different history than peridotites from the southern (lherzolitic) SCLM domain. Strong influence of the SSZ-related melts is pronounced for harzburgite rim spinels and dunite resorbed spinels. The composition of the core changes toward the SSZ-modified compositions that can be indicative of disruption of these lithologies by the SSZ-related melts.

The MORB-like melts influenced all studied peridotites and could be related to the development of the Lambert–Amery rift system from 390–320 Ma [30,46,47]. The features obtained by spinels due to the MORB-like melts influence could be related to stage three of the Lambert–Amery rift system development (after [4]) and may be associated with the beginning of intensive tholeiite dyke magmatism (320 Ma; [4,46]). The SSZ-related melts influenced the deepest sampled levels of the southern upper mantle domain and almost the whole upper mantle column of the northern domain might be related to reactivation of SSZ slab material (buried at the deeper levels of the upper mantle during amalgamation of East Antarctica either ca. 1400–1000 Ma or ca. 580–500 Ma; see [6,13] and references therein). The precise timing of such reactivation is not clear yet, but it could have happened between the infiltration of the MORB-like melts incorporation of the xenoliths to the ascending host magma (150–140 Ma [1]). The reactivation could take place, for example, during stage four of the Lambert–Amery rift system development (after [4]) and have occurred within 2 Ma before the host alkaline-ultramafic magmas emplacement (see [4,45]).

## 5. Conclusions

Two Late Jurassic to Early Cretaceous intrusions in the Jetty Peninsula of the East Antarctica sampled two different SCLM domains (one mostly lherzolitic and another one non-lherzolitic). Modal abundance of Cr-spinels in the studied peridotites is dunite > Cpx-poor lherzolite ≥ Cpx-rich lherzolite ≥ harzburgite > Spl-Grt lherzolite. Three main textural types of spinel were identified: (i) primary spinel represented by clean homogeneous grains, (ii) a rim of recrystallization/resorption surrounding primary spinel grains and (iii) irregular interstitial resorbed grains. Most trace elements analyzed in spinels are present in very low amounts. Only Ti, V, Mn, Co, Ni, Zn and Ga display concentrations in the range of tens to hundreds (up to thousands) ppm. Other trace element

concentrations vary from below the detection limit to <10 ppm. Major oxide and trace element features of the studied spinels provided information on the processes of depletion, enrichment and melt to rock interaction in the lithospheric upper mantle beneath the region. The two sampled SCLM domains were differently affected by the processes of partial melting and following metasomatic enrichment. The upper mantle peridotites represent both simple melt residues (Cpx-poor Spl lherzolites from the Yuzhnoe intrusion) and residues strongly influenced by the melts migrating through the peridotite matrix. These melts are the MORB-like melts (affecting Cpx-rich Spl and Spl-Grt lherzolites from the Yuzhnoe intrusion) likely related to the development of the Lambert–Amery rift system and the SSZ-related melts (affecting non-lherzolite peridotites from the Severnoe intrusion and a part of Spl-Grt lherzolites from the Yuzhnoe intrusion) likely related to reactivation of the SSZ material buried during the amalgamation of East Antarctica. The MORB-like melts influenced all studied peridotites to different extents, whereas the SSZ-related melts influenced the deepest sampled levels of the southern upper mantle domain and almost the whole upper mantle column of the northern domain. Some secondary spinels from dunite xenoliths might have crystallized directly from the melt portions infiltrating through the peridotite matrix.

**Supplementary Materials:** The following supporting information can be downloaded at: https://www.mdpi.com/xxx/s1, Table S1: Major oxide and mineral compositions of spinels in upper mantle xenoliths from Jetty Peninsula (all data); Table S2: Trace element compositions of spinels in upper mantle xenoliths from Jetty Peninsula (all data).

**Author Contributions:** Conceptualization, A.V.A. and I.E.A.; methodology, I.E.A.; investigation, I.E.A., A.V.A. and O.P.; resources, I.E.A. and O.P.; writing, A.V.A. and I.E.A.; funding acquisition, I.E.A. All authors have read and agreed to the published version of the manuscript.

**Funding:** The research was funded by the Czech Geological Survey (project 311250 to I.E.A.).

**Data Availability Statement:** The data supporting the findings of this study are available within the article and its supplementary materials.

**Acknowledgments:** We are indebted to the late Eugeny Mikhalsky (1956–2021) and Anatoly Laiba (1954–2016), outstanding Antarctic geologists and petrologists, to whose memory we dedicate this contribution. The authors thank T. Sidorinova for help with optical microscopy. Four anonymous reviewers, I. Ashchepkov and the academic editor are thanked for their comments and suggestions. This is a contribution to the Research Project 311250 which is a part of the Strategic Research Plan of the Czech Geological Survey (DKRVO/CGS 2018–2022).

**Conflicts of Interest:** The authors declare no conflict of interest. The funders had no role in the design of the study; in the collection, analyses, or interpretation of data; in the writing of the manuscript; or in the decision to publish the results.

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
