# Peer review of "Major and Trace-Element Chemistry of Cr-Spinel in Upper Mantle Xenoliths from East Antarctica"

_minerals, doi:10.3390/min12060720_

Round 1

Reviewer 1 Report

This paper reports a detailed and well-designed petrographic study of Cr-spinels in the upper mantle peridotite xenoliths from two intrusions of alkaline-ultramafic rocks in Jetty Peninsula

The study is undoubtedly a useful contribution and I have only minor comments for improvement.

I hope the comments and suggestions are useful to bring this manuscript to the level where it can be published in Minerals.

1.     It is not clear from the methodological part of the manuscript, how many analyses have you done per one grain? Did you check mineral zoning in the primary spinels?

2.     In the part 3.1.4 You discuss composition of the “porous” Spl, however in the beginning of the chapter you give only brief information about the third type – resorbed spinel without any comments about the “porous” texture. Information about the porous material of the spinel is supported only by the Fig. Maybe you should give more information in the text, also in the discussion part about the genetic aspects.

3.     Taking the uncertainties into account composition of the primary spinel and rim is identical (except for Fe)

4.     How did you calculate the percentages for partial melting for the host peridotite for the Yuzhnoe body and Severnoe body? Have you used similar equations as in Ref 34?

5.     Fig. 5b is too small for good presentation of different types of spinel.

6.     Fig. 7. Did you analyze the interstitial “glass” in the spinel? Is it isotropic? You have to prove that it is not the secondary minerals (serpentine?). For example, by Raman spectroscopy you can analyze this phase. Also, Figure 7 is not representative to show two types of spinel.

7.     «Spinel rims from harzburgite and dunite xenoliths are characterized by erratically higher TiO2 contents at slightly higher Cr# compared to primary spinels (Figure 13)» - Dunite spinels does not show any variation in TiO2 according to Fig. 13.

8.      Text contain typos that should be carefully corrected. For example “Cpx poor Spl herzolites toward Spl-Grt lherzolites”

Author Response

This paper reports a detailed and well-designed petrographic study of Cr-spinels in the upper mantle peridotite xenoliths from two intrusions of alkaline-ultramafic rocks in Jetty Peninsula

The study is undoubtedly a useful contribution and I have only minor comments for improvement.

I hope the comments and suggestions are useful to bring this manuscript to the level where it can be published in Minerals.

  1. It is not clear from the methodological part of the manuscript, how many analyses have you done per one grain? Did you check mineral zoning in the primary spinels?

A - Mostly, we did two spots per sample: at the center of the grain and at the rim (this information is added to the methodological part). The difference in the composition was significant. On the other hand, when we analyzed two spots for the rim-free spinels (the core and the edge), there were no significant compositional differences. Unfortunately, we were very limited in the financial support, so we had to try to minimize the number of individual analyses. However, a detailed study by Foley et al. (2006; [4]) showed that Jetty Peninsula peridotite spinels do not display major oxide zonation beyond the compositional difference between the recrystallization rim and the rest of the grain.

  1. In the part 3.1.4 You discuss composition of the “porous” Spl, however in the beginning of the chapter you give only brief information about the third type – resorbed spinel without any comments about the “porous” texture. Information about the porous material of the spinel is supported only by the Fig. Maybe you should give more information in the text, also in the discussion part about the genetic aspects.

A – The term “porous” has mistakenly left from the earlier versions of the manuscript when we were looking for the term reflecting the features of this spinel type to the best. At that earlier stage we called the completely resorbed grains “porous”. Later on, we omitted this term. The correction is made in the text and in Figure 6.

  1. Taking the uncertainties into account composition of the primary spinel and rim is identical (except for Fe)

A – I am sorry, but we would have to disagree with you. We guess that you made the conclusions (rim vs. core) just from the descriptive part of the manuscript. But there, we have just given the total ranges of the rim and core compositions. If you have a look at the Table 1 and Table S1, you would see that the each pair “core – corresponding rim” have significantly different concentrations of the most characteristic elements such as Al and Cr. This is also pronounced in the Cr# of the core and the rim.

  1. How did you calculate the percentages for partial melting for the host peridotite for the Yuzhnoe body and Severnoe body? Have you used similar equations as in Ref 34?

A – Yes, we used the similar equation as in Ref 34 (now, Ref 36).

  1. Fig. 5b is too small for good presentation of different types of spinel.

A – All spinel photos are also given as high-resolution images (Electronic supplement Figures S1), which will be available in the published version of the manuscript through a simple click on the picture: it is a big advantage of existence of the electronic version of the paper.

  1. Fig. 7. Did you analyze the interstitial “glass” in the spinel? Is it isotropic? You have to prove that it is not the secondary minerals (serpentine?). For example, by Raman spectroscopy you can analyze this phase. Also, Figure 7 is not representative to show two types of spinel.

A – Yes, we analyzed glass in harzburgite SN-N18 spinel (just two analyzes), and the results are given in the text (the 3rd paragraph of the Discussion, just before the section 4.1). The matter is isotropic, but we did not applied any special analytical method to this material. First of all, because the study of the glass was beyond the scope of the present study (we just provided the analyses for informative reasons), and the second – the project had a very low budget, and we simply did not have money for any extended additional study. As a matter of fact, similar glasses were described by Foley et al. (2006; [4]) and Andronikov (1997; [29]) for the same xenolith suite.

And, with all due respect, Figure 7 demonstrates various types of spinel very good. I am not even talking about Figures 7a,c,d,e where we can see primary solid homogeneous spinel (7a,d), completely resorbed spinel grains (7c), and secondary, so-called “skeletal” grains (7e), but also in Figure 7b one can see primary homogeneous spinel core (Spl-1) and the rim (Spl-2) which just started to develop. It is just necessary to have a look at Figures in the high resolution (they are provided as separate files “Figure S1” during the submission).

  1. «Spinel rims from harzburgite and dunite xenoliths are characterized by erratically higher TiO2 contents at slightly higher Cr# compared to primary spinels (Figure 13)» - Dunite spinels does not show any variation in TiO2 according to Fig. 13.

A – You are absolutely right. “Dunite” is deleted from this sentence. It is completely our fault. “Dunite” has likely left in the text from some of the earliest versions of the manuscript.

  1. Text contain typos that should be carefully corrected. For example “Cpx poor Spl herzolites toward Spl-Grt lherzolites”

A – We tried to do our best in order to check and correct all the typos.

Reviewer 2 Report

Dear authors

You can find the minor spell checks and suggestions in the attached file.

I strongly suggest you put some legend in the figures, besides, it would gave much more attention to the reader, if you add some phrases to link geochemical meaning of the data to the overall tectonics of the region

best regards

Author Response

Dear authors

You can find the minor spell checks and suggestions in the attached file.

A – We tried to do our best in order to check and correct all the typos. And thank you very much for pointing us at some of such typos.

I strongly suggest you put some legend in the figures, besides, it would gave much more attention to the reader, if you add some phrases to link geochemical meaning of the data to the overall tectonics of the region.

A – We modified Figure 9 inserting most legend to the Figure body. Figure Caption was modified accordingly. We have added some phrases about tectonics at the end of the section 4.3. But generally, it is difficult to link certain geochemical Spl features with overall tectonics of the region without a complex comprehensive study of everyting.

Reviewer 3 Report

The paper Trace-Element Geochemistry of Cr-Spinel in the Upper Mantle Xenoliths from East Antarctica

After Alexandre V. Andronikov * , Irina E Andronikova , Ondrej Pour

The paper produce  very good  impression. The topic is very interesting because the object such as mantle  xenoliths from Antartica are really rare and they are a bitt differ from the similar Sp-Ga xenoliths in alkali basalts Worldwide.  

The ICP analyses of the Cr- spinels are also rather rare.

Of course  the xenoliths were described several time  in the scientific  literature but the trace elements in mineral  describe not in detail.  

All descriptions of methods and petrography of xenoliths and texture are in very high level/

Authors found the difference in trace elements  between the groups estimated according petrography and textural occurrence. The re - fertilization by the partial melts initiated by the plume is the major factor of the evolutions like it was determined in  many  case for mantle  undergone to the plume magmatism

What are not clear things. Of course PT-FO2 conditions. Though  they were described in previous paper but here they at least  should  be mentioned.

Also it is not clear why mainly elements of siderophile and chalcophile elements were used in the variation diagrams.  The high field strengthened elements (not only Ti) should ve presented in diagrams also.

The interesting question – the rather high concentration of the REE elements with the inclination similar to alkaline basalts (or peridotite partial melts). And what is the reason  - microinclusions or primary features of spinels.  

In the diagrams it is better to give the legend with the symbols because the description needs some time to understand.

Paper could be published after  minor revision

Best wishes Igor Ashchepkov

Author Response

The paper produce  very good  impression. The topic is very interesting because the object such as mantle  xenoliths from Antartica are really rare and they are a bitt differ from the similar Sp-Ga xenoliths in alkali basalts Worldwide.  

The ICP analyses of the Cr- spinels are also rather rare.

Of course  the xenoliths were described several time  in the scientific  literature but the trace elements in mineral  describe not in detail.  

All descriptions of methods and petrography of xenoliths and texture are in very high level/

Authors found the difference in trace elements  between the groups estimated according petrography and textural occurrence. The re - fertilization by the partial melts initiated by the plume is the major factor of the evolutions like it was determined in  many  case for mantle  undergone to the plume magmatism

What are not clear things.

Of course PT-FO2 conditions. Though they were described in previous paper but here they at least  should  be mentioned.

A – As a matter of fact, this information is provided at the end of the section 2.1. Since we did not do our own calculations, this information is taken from the previously published works ([3, 4, 8]).

Also it is not clear why mainly elements of siderophile and chalcophile elements were used in the variation diagrams.  The high field strengthened elements (not only Ti) should ve presented in diagrams also.

A – Unfortunately, most trace elements are present in spinels at the very low levels (often very close to the detection limits of the methods). Therefore, inclusion of these elements in various diagrams would not make any sense. Actually, such diagrams would be very useful in the case of spinels from mantle xenoliths found in the West Antarctica ([31]), which xenoliths were the subject of much more intensive metasomatic influence. But unfortunately, these xenoliths do not belong to our collection.

The interesting question – the rather high concentration of the REE elements with the inclination similar to alkaline basalts (or peridotite partial melts). And what is the reason  - microinclusions or primary features of spinels.

A – With all due respect, the concentrations of the REE are very low in the spinels studied (which is different from spinels from the West Antarctica [31]). The high REE concentrations (when occur in the spinels studied) are due incorporation of silicate melts in the spinel porous matter that compromises composition spinels. This phenomenon is described in the text of the manuscript and is demonstrated by the LA-ICP-MS depth profile in Figure 8.

In the diagrams it is better to give the legend with the symbols because the description needs some time to understand.

A – Actually, we have already provided the Legend within the Figure 9 following an advise of one of the reviewer, and this alone strongly overloaded the Figure, although such a Legend is obviously necessary. However, if we would add the Legend to each Figure, the Figures will become compltely unreadable. So, we would like to keep the whole Legend only in Figure 9, and just refer in other figures to that Legend.

Reviewer 4 Report

The manuscript “Trace-Element Geochemistry of Cr-Spinel in the Upper Mantle Xenoliths from East Antarctica” studied the major and trace-element compositions by SEM and LA-ICP-MS of the Cr-spinels in the upper mantle peridotite xenoliths from two Late Mesozoic intrusions of alkaline-ultramafic rocks in Jetty Peninsula (East Antarctica) in order to discuss their petrogenesis. Τhe results are well presented. The authors have employed correctly the techniques. I have found the methodological approach correct. The presentation of the problem is clear, the results correctly presented and the conclusions well explained. English is not bad and generally is easy to follow, but there are some evident grammar mistakes. To conclude, I suggest this manuscript to be published in the journal “Minerals” after the below minor revisions:

Ø  The first section introduces the problem to a reader. It is well written, concise and informative. Some references should be added in the introduction for the spinel-group mineral composition.

References to be cited in the introduction field:

·       Petrogenetic Implications for Ophiolite Ultramafic Bodies from Lokris and Beotia (Central Greece) Based on chemistry of their Cr-spinels. Geosciences 2017, 7, 10.

·       Mineralogical Evidence for Partial Melting and Melt-Rock Interaction Processes in the Mantle Peridotites of Edessa Ophiolite (North Greece). Minerals 2019, 9, 120.

Ø  The purpose of the study needs better wording.

Ø  Please provide a detailed description for the geological characteristics of the studied area.

Ø  Fig. 1: Do you have a sampling map of where more precisely the samples are from?

Author Response

The manuscript “Trace-Element Geochemistry of Cr-Spinel in the Upper Mantle Xenoliths from East Antarctica” studied the major and trace-element compositions by SEM and LA-ICP-MS of the Cr-spinels in the upper mantle peridotite xenoliths from two Late Mesozoic intrusions of alkaline-ultramafic rocks in Jetty Peninsula (East Antarctica) in order to discuss their petrogenesis. Τhe results are well presented. The authors have employed correctly the techniques. I have found the methodological approach correct. The presentation of the problem is clear, the results correctly presented and the conclusions well explained. English is not bad and generally is easy to follow, but there are some evident grammar mistakes. To conclude, I suggest this manuscript to be published in the journal “Minerals” after the below minor revisions:

Ø  The first section introduces the problem to a reader. It is well written, concise and informative. Some references should be added in the introduction for the spinel-group mineral composition.

References to be cited in the introduction field:

  • Petrogenetic Implications for Ophiolite Ultramafic Bodies from Lokris and Beotia (Central Greece) Based on chemistry of their Cr-spinels. Geosciences 2017, 7, 10.
  • Mineralogical Evidence for Partial Melting and Melt-Rock Interaction Processes in the Mantle Peridotites of Edessa Ophiolite (North Greece). Minerals 2019, 9, 120.

A – All recommended references were added to the text (both in the Introduction and in the Discussion) and to the Reference list (new Refs. [18 and 19])

Ø  The purpose of the study needs better wording.

A – We tried to paraphrase this section.

Ø  Please provide a detailed description for the geological characteristics of the studied area.

A – This is a sheer mineralogical-geochemical work, and we did not conduct any special geological study. The detailed characteristics of the geology of the region is given in multiple publications elsewhere (see [1, 4, 6, 9-13] and in other related references in the list). We added, however a sentence to the text which briefly explains the tectonic setting of the studied area. We strongly believe that that would be enough for the purposes of the current work.

Ø  Fig. 1: Do you have a sampling map of where more precisely the samples are from?

A – As a matter of fact, we tried to provide such a map for our paper dealing with sulfides (Andronikov et al., 2021; [7] in the References). However, the question about copyrights has immediately arisen (since we do not have a map drawn by our own, and could only copy already published maps). Therefore, we decided just to refer to already published works [4 and especially 6], where all necessary maps are provided in great details. Nevertheless, we added a phrase referring these works to the Caption for Figure 1. Actually, we believe that since this work totally deals with mineralogy and geochemistry, any detailed pictures for structure of the region as a whole or intrusive bodies as the units are not vitally necessary for this particular work.

Reviewer 5 Report

I have completed a review of the manuscript entitled:

Trace-Element Geochemistry of Cr-Spinel in the Upper Mantle Xenoliths from East Antarctica

The petrology and mineral chemistry are welcome additions to the little currently known about Antarctic mantle xenolith chemistry. The data are well presented in tables in the manuscript and in the online supplementary material. The petrology photographs are clear and well annotated and the chemical discrimination diagrams interesting, though I have made recommendations for reducing and consolidating their number. The text and arguments presented are mostly clear, I have made a few suggestions below but the manuscript could do with careful checking before final submission. Below are a few major and minor comments. I look forward to seeing this published.

Regards,  

Major comments:

I suggest referring to ‘major’ as well as trace elements in your title, as the majority of the results and plots use the major element data. E.g. Major and trace element chemistry of Cr-spinel in mantle xenoliths from East Antarctica

I would include a legend in each of your figures, as they aren’t really stand alone without that.

I would reduce your numbers of figures. You might do this by i. amalgamating some of the petrology plates ii. Moving the depth profile figures to a supplementary file, iii. Better combining the geochemistry plots  

Please describe how you deal with iron throughout the manuscript. I assume it is assigned into Fe2+/3+ using Droop 87 or similar, but this should be described.

Minor comments

Section 1

-          paragraph 2, remove plural ‘s’ to detail (c.f. details)

-          Another Antarctic spinel trace element chemistry source is the supplementary file in:

-          See supplementary material for spinel trace element chemistry from south Victoria Land Martin, A.P.; Cooper, A.F.; Price, R.C.; Doherty, C.L.; Gamble, J.A. 2021 A review of mantle xenoliths in volcanic rocks from southern Victoria Land, Antarctica. doi: 10.1144/M56-2019-42 IN: Martin, A.P.; van der Wal, W. (eds) The geochemistry and geophysics of the Antarctic mantle. London: Geological Society. Memoir (Geological Society of London) 56

-          LA-ICP-MS should be changed to spectrometry (c.f. spectrometric)

Section 2.1

-          change at% to wt%

-          Remove word ‘massive’ in first sentence describing dunites.

-          You can probably add phlogopite as an example of metasomatic enrichment

-          “sulfide blebs occurring as inclusions in spinels” this is interesting, so you have oikocrysts of spinel poikiliticlly enclosing chadocrysts of sulphide? You can say something here about order of mineral growth and re-equilibration I suggest

-          Spelling of figacity

Section 2.2.1

-          again use ‘spectroscopy’, c.f. spectroscopic

Section 2.2.2

-          “An internal standardization was based on Fe concentrations determined by the SEM analysis” – you need to mention how iron was treated. Was it all treated as FeO? Or was there a calculation of Fe2+ versus Fe3+? This should be stated here or in the previous section. I see you have Fe2+/3+ in the online supplement, this looks good. Please include the method you used to calculate this in this section, e.g. Droop ’87, etc.

-          Please list all of your ’43 isotopes monitored during the analyses please. I count 30 + REE – and I think there should be 15 REE.

Section 3.1.1

-          spelling if ‘reprsented’

Section 3.1.3

-          Please spell ulvöspinel correctly, c.f. ulvospinel  

Section 5.1.5

-          could you please reference a figure photograph showing the spinel ‘forming grain clusters’?

Section 3.2

-          explain acronym ppm at first use

Section 4.1.

-          Undo delete on “the Yuzhnoe intrusion” because the sentence needs a subject

-          Define MORB at first use in text

-          I recommend sticking with MORB-like rather than MOR-like, for consistency. Or if you do use both, please explain each at first usage

-          Is ‘sheer melting’ the correct term here? Maybe just ‘melting’ (by whatever method)

-          Explain SSZ at first use, though there is nothing wrong with using suprasubduction zone-like melts without another abbreviation introduced into the manbuscript in my opinion

-          hot silicate melts – I recommend deleting the qualitative word ‘hot’ as you give no value for it and I suggest the temperature is unknown

-          Fix spelling of compoaitions

Section 4.2

-          Fix spelling of display

-          Define OIB at first usage in text

Figures / tables

-          Figure 1. The symbol is a star not an asterisk (spelling). Give abbreviation for Mtns. Or spell out.

-          Figure 4 (and elsewhere). Is ‘structural’ the right word here? I think you could delete this word from the entire manuscript and improve the clarity of your descriptions.

-          Figure 9. Bring the symbol level above the green MORB field. The reason for this is the Spl rim; Spl-Grt lherzolite symbol is black in the legend but green on the figure. Spell out all abbreviations in the caption.

-          Figure 12. Change caption to read: Plots of mantle spinel Cr/Al vs. various trace elements and Ni/Co from East Antarctic xenoliths.  Note, include units for all y-axis elements (ppm?), but not Ni/Co

-          Figure 13. I would normally list the x-axis value 1st, i.e. TiO2 (wt%) vs. Cr#, by convention. Explain all acronyms in the caption.

-          Figure 14, 15. Explain all acronyms in the caption.

-          Table 1. Make oxide numerals lower case in far-left column. By convention, ALL abbreviations should be spelled out as a footnote to all tables

Supplement

-          Are the blanks in Table S1 (majors) below the lower method detection limit? Or nat analysed? I recommend filling this out as it will help other researchers.

-          I would re-include your definitions of Cr#, Mg# and Fmelt in the tables

Author Response

The petrology and mineral chemistry are welcome additions to the little currently known about Antarctic mantle xenolith chemistry. The data are well presented in tables in the manuscript and in the online supplementary material. The petrology photographs are clear and well annotated and the chemical discrimination diagrams interesting, though I have made recommendations for reducing and consolidating their number. The text and arguments presented are mostly clear, I have made a few suggestions below but the manuscript could do with careful checking before final submission. Below are a few major and minor comments. I look forward to seeing this published.

Regards,  

Major comments:

I suggest referring to ‘major’ as well as trace elements in your title, as the majority of the results and plots use the major element data. E.g. Major and trace element chemistry of Cr-spinel in mantle xenoliths from East Antarctica

A – Yes, we agree, and have changed the title accordingly

I would include a legend in each of your figures, as they aren’t really stand alone without that.

A – With all due respect, I would not agrre with the reviewer. We have already provided the Legend within the Figure 9 following an advise of one of the reviewer, and this alone strongly overloaded the Figure, although such a Legend is obviously necessary. However, if we would add the Legend to each Figure, they will become compltely unreadable. So, we would like to keep the whole Legend only in Figure 9, and just refer in other figures to that Legend.

I would reduce your numbers of figures. You might do this by i. amalgamating some of the petrology plates ii. Moving the depth profile figures to a supplementary file, iii. Better combining the geochemistry plots 

A – I would not like to tell that from my personal experience, it is always a headace to search for the data using Supplementary Files available somewhere in the depth of the internet (maybe I am a little bit old fashioned guy). I have always prefered to see all the information directly in the body of the paper. I guess, the only data which should really be given as Suppementary Files are very long tables and some raw data, like those in isotopic geochronology, etc. Therefore, if it is not against the Journals‘ rules,  I would prefer to keep all the Figures there where they are now. I think, it is a big advantage of journals which are issued in the electronic form only, and are not restricted by the volume of the printed pages. As for combining a few plates in one, I would not do that because figures are mostly small with the size just enough for the information to be comprehended, and further reduction of the figures size would make them almost unreadable.

Please describe how you deal with iron throughout the manuscript. I assume it is assigned into Fe2+/3+ using Droop 87 or similar, but this should be described.

A – Yes, you are absolutely correct: we used a standard equation from Droop (1987; it is now [29] in the reference list) based on stoichiometric criteria. Obviously, we did not invent anything new.

Minor comments

Section 1

-          paragraph 2, remove plural ‘s’ to detail (c.f. details)

A - Done

-          Another Antarctic spinel trace element chemistry source is the supplementary file in:

        See supplementary material for spinel trace element chemistry from south Victoria Land Martin, A.P.; Cooper, A.F.; Price, R.C.; Doherty, C.L.; Gamble, J.A. 2021 A review of mantle xenoliths in volcanic rocks from southern Victoria Land, Antarctica. doi: 10.1144/M56-2019-42 IN: Martin, A.P.; van der Wal, W. (eds) The geochemistry and geophysics of the Antarctic mantle. London: Geological Society. Memoir (Geological Society of London) 56

A – Yes, thank you! From now on, we refer to this paper in our work [31].

-          LA-ICP-MS should be changed to spectrometry (c.f. spectrometric)

A – Done

Section 2.1

-          change at% to wt%

A - Done

-          Remove word ‘massive’ in first sentence describing dunites.

A – Done

-          You can probably add phlogopite as an example of metasomatic enrichment

A - The phlogopite is added to the description of dunites

-          “sulfide blebs occurring as inclusions in spinels” this is interesting, so you have oikocrysts of spinel poikiliticlly enclosing chadocrysts of sulphide? You can say something here about order of mineral growth and re-equilibration I suggest

A – Yes,this is interesting, but we have already described in detail most features of sulfides from the same xenolith suite (Andronikov et al. (2021; [7]).

-          Spelling of figacity

A - Corrected

Section 2.2.1

-          again use ‘spectroscopy’, c.f. spectroscopic

A - Corrected

Section 2.2.2

-          “An internal standardization was based on Fe concentrations determined by the SEM analysis” – you need to mention how iron was treated. Was it all treated as FeO? Or was there a calculation of Fe2+ versus Fe3+? This should be stated here or in the previous section. I see you have Fe2+/3+ in the online supplement, this looks good. Please include the method you used to calculate this in this section, e.g. Droop ’87, etc.

A – Yes, we have included Droop (1987; [29]) in both the text and the reference list.

-          Please list all of your ’43 isotopes monitored during the analyses please. I count 30 + REE – and I think there should be 15 REE.

A – Yes, right, it was our fault. We measured 14 REE isotopes (obviously, we did not measure Pm). So, alltogether there were 44 isotopes. But now, we are just listing all the isotopes measured.

Section 3.1.1

-          spelling if ‘reprsented’

A – Corrected

Section 3.1.3

-          Please spell ulvöspinel correctly, c.f. ulvospinel  

A – Corrected, both in the text and tables.

Section 3.1.5

-          could you please reference a figure photograph showing the spinel ‘forming grain clusters’?

A – Frankly speaking, this has left from one of the earlier versions of the manuscript when we were searching for proper names and terms. In reality, no clsuters were observed. Spinels are just unevenly scattered through the rock’s body. Correction is made in the text.

Section 3.2

-          explain acronym ppm at first use

A – Done, although, I do not think it is really necessary, since this is a commonly used acronym.

Section 4.1.

-          Undo delete on “the Yuzhnoe intrusion” because the sentence needs a subject

A – Corrected. Sorry, it was completely our fault.

-          Define MORB at first use in text

A – Done

-          I recommend sticking with MORB-like rather than MOR-like, for consistency. Or if you do use both, please explain each at first usage

A – Yes, we would follow your advise to use the term MORB for the sake of consistency.

-          Is ‘sheer melting’ the correct term here? Maybe just ‘melting’ (by whatever method)

A – Yes, you are right, just “melting“.

-          Explain SSZ at first use, though there is nothing wrong with using suprasubduction zone-like melts without another abbreviation introduced into the manuscript in my opinion

A – SSZ is explained.

-          hot silicate melts – I recommend deleting the qualitative word ‘hot’ as you give no value for it and I suggest the temperature is unknown

A – Corrected

-          Fix spelling of compoaitions

A – Done

Section 4.2

-          Fix spelling of display

A – Done

-          Define OIB at first usage in text

A - Done

Figures / tables

-          Figure 1. The symbol is a star not an asterisk (spelling). Give abbreviation for Mtns. Or spell out.

A – Corrected

-          Figure 4 (and elsewhere). Is ‘structural’ the right word here? I think you could delete this word from the entire manuscript and improve the clarity of your descriptions.

A – “Structural“ is deleted.

-          Figure 9. Bring the symbol level above the green MORB field. The reason for this is the Spl rim; Spl-Grt lherzolite symbol is black in the legend but green on the figure. Spell out all abbreviations in the caption.

A – Corrected

-          Figure 12. Change caption to read: Plots of mantle spinel Cr/Al vs. various trace elements and Ni/Co from East Antarctic xenoliths.  Note, include units for all y-axis elements (ppm?), but not Ni/Co

A – Corrected according to your advise.

-          Figure 13. I would normally list the x-axis value 1st, i.e. TiO2 (wt%) vs. Cr#, by convention. Explain all acronyms in the caption.

A – Corrected for all appropriate diagrams, and also in the text.

-          Figure 14, 15. Explain all acronyms in the caption.

A – All acronyms are explained

-          Table 1. Make oxide numerals lower case in far-left column. By convention, ALL abbreviations should be spelled out as a footnote to all tables

A – Low cases are made. Abbreviations are spelled out.

Supplement

-          Are the blanks in Table S1 (majors) below the lower method detection limit? Or not analysed? I recommend filling this out as it will help other researchers.

A – The “bdl“ abbreviation is added to the blanks and is expalined in the Table’s notes.

-          I would re-include your definitions of Cr#, Mg# and Fmelt in the tables

A - Done
